# MoLE 🏖️: Enhancing Human-centric Text-to-image Diffusion via **Mixture of Low-rank Experts**

**Jie Zhu**[1,2], **Yixiong Chen**[3], **Mingyu Ding**[4], **Ping Luo**[5], **Leye Wang**[1,2*], **Jingdong Wang**[6*]

[1]Key Lab of High Confidence Software Technologies (Peking University), Ministry of Education, China
[2]School of Computer Science, Peking University, Beijing, China, [3]Johns Hopkins University
[4]UC Berkeley, [5]The University of Hong Kong, [6]Baidu
zhujie@stu.pku.edu.cn, ychen646@jh.edu, myding@berkeley.edu,
pluo@cs.hku.hk, leyewang@pku.edu.cn, wangjingdong@outlook.com

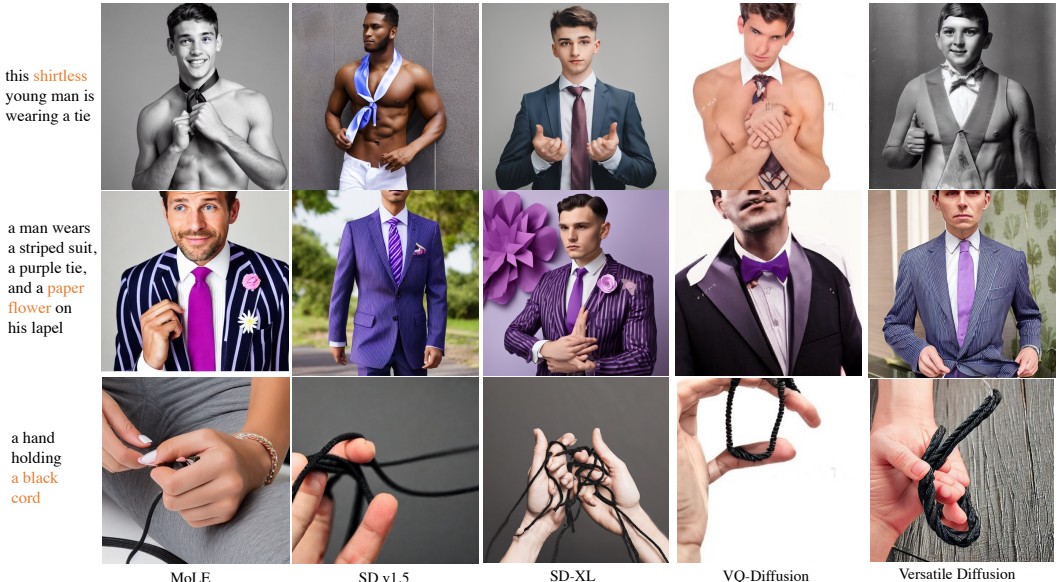

this shirtless young man is wearing a tie

a man wears a striped suit, a purple tie, and a paper flower on his lapel

a hand holding a black cord

MoLE    SD v1.5    SD-XL    VQ-Diffusion    Versatile Diffusion

Figure 1: Compare MoLE with other diffusion models. Pay more attention to the face and (especially) hand. Zoom in for a better view.

## Abstract

Text-to-image diffusion has attracted vast attention due to its impressive image-generation capabilities. However, when it comes to human-centric text-to-image generation, particularly in the context of faces and hands, the results often fall short of naturalness due to insufficient training priors. We alleviate the issue in this work from two perspectives. **1)** From the data aspect, we carefully collect a *human-centric dataset* comprising over one million high-quality human-in-the-scene images and two specific sets of close-up images of faces and hands. These datasets collectively provide a rich prior knowledge base to enhance the human-centric image generation capabilities of the diffusion model. **2)** On the methodological front, we propose a simple yet effective method called **M**ixture **o**f **L**ow-rank **E**xperts (**MoLE**) by considering low-rank modules trained on close-up hand and face images respectively as experts. This concept draws inspiration from our observation of low-rank refinement, where a low-rank module trained by a customized close-up dataset has the potential to enhance the corresponding image part when applied at an appropriate scale. To validate the superiority of MoLE in the context of human-centric image generation compared to state-of-the-art, we

---

[*]Corresponding author

38th Conference on Neural Information Processing Systems (NeurIPS 2024).

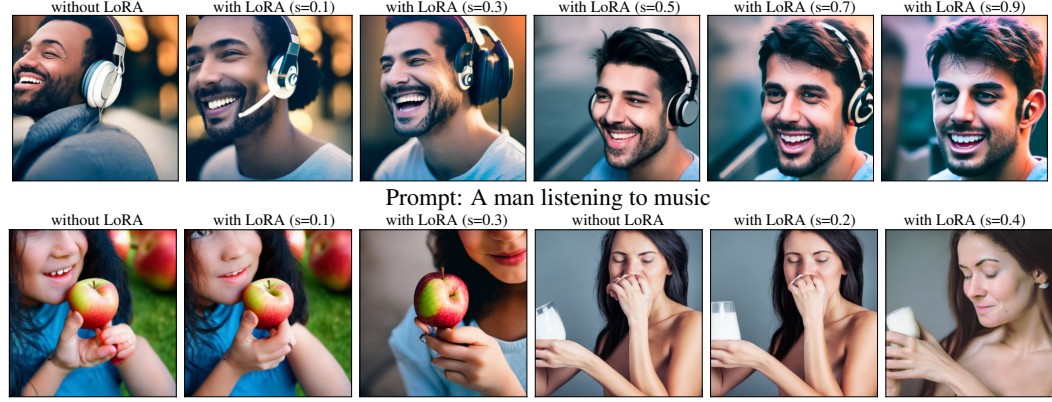

| without LoRA | with LoRA (s=0.1) | with LoRA (s=0.3) | with LoRA (s=0.5) | with LoRA (s=0.7) | with LoRA (s=0.9) |

Prompt: A man listening to music

| without LoRA | with LoRA (s=0.1) | with LoRA (s=0.3) | without LoRA | with LoRA (s=0.2) | with LoRA (s=0.4) |

Prompt: A girl eating an apple          Prompt: A woman drinking milk

Figure 2: **Inspiration of Mixture of Low-rank Experts.** In the first and second row, we train two low-rank modules on SD v1.5 simply using off-the-shelf Celeb-HQ face dataset [22] and 11k Hands dataset [1], respectively. *With a proper scale weight, low-rank module can refine corresponding part.* We term this phenomenon as low-rank refinement.

construct two benchmarks and perform evaluations with diverse metrics and human studies. Datasets, model, and code are released at project website.

# 1 Introduction

Human-centric text-to-image generation is an important orientation for realistic applications, *e.g.*, poster design, virtual reality, *etc*. However, current models encounter issues with producing natural-looking results, particularly in the context of faces and hands [2]. To address this issue, we delve into the matter and identify two factors that may contribute to this issue. Firstly, the absence of *high-quality* human-centric data makes diffusion models lack sufficient human-centric prior [3]; Secondly, in the human-centric context, faces and hands represent the two most complicated parts due to high variability, making them challenging to be generated naturally.

Hence, we alleviate this problem from two perspectives. On the one hand, we collect a human-in-the-scene dataset of high quality and high resolution from the Internet. Basically, the resolution of each image is various and over $1024 \times 2048$. The dataset contains approximately one million images, and covers different races, various gestures, and activities, thereby providing diffusion models with sufficient knowledge to improve the performance of human-centric generation. However, for the second factor, our experiment in Sec 5.3 demonstrates though fine-tuning on above dataset brings an overall enhancement in human quality, human face and hand still exhibit unnatural outcomes, possibly because diffusion models focus more on overall performance during fine-tuning while struggling to accurately capture highly variable parts like face and hand gestures.

To address the second factor, an interesting *low-rank refinement phenomenon* inspires us. As shown in Fig 2, when combined with a customized low-rank module [19] and using a proper scale weight, Stable Diffusion v1.5 (SD v1.5) [40] has the potential to refine the corresponding part of a person, *e.g.*, for hand, $s = 0.4$ subtly refines the appearance of the woman drinking milk's hand. Thus, inspired by this, to refine the face and hand, we can first gather two customized high-quality datasets (one for face close-ups and one for hand close-ups) to train two low-rank modules, respectively. Then, for the two specialized low-rank modules, we could add a certain assignment to adaptively select which low-rank module to use for a given input and Mixture of Experts (MoE) naturally stands out. Moreover, as face and hand often appear simultaneously in an image for a person, motivated by Soft MoE [35], we could adopt a soft assignment to produce adaptive scale weights, activating multiple experts to handle the input at the same time. We refer to all mentioned three datasets above together as *human-centric dataset* for convenience as shown in Fig 3.

---

[2] The HuggingFace website acknowledges that "Faces and people in general may not be generated properly." Despite some efforts, such as those seen in DALL-E 3 and Midjourney, aimed at addressing this issue, they are generally closed-sourced for business. Our objective is to establish transparent and open-source endeavors for the advancement of the community in generating more realistic human hands/faces.

[3] We sample 350k human-centric images from LAION2b-en and the average height and width are 455 and 415, among which most are between 320 and 357, limited in providing sufficient human-centric knowledge.

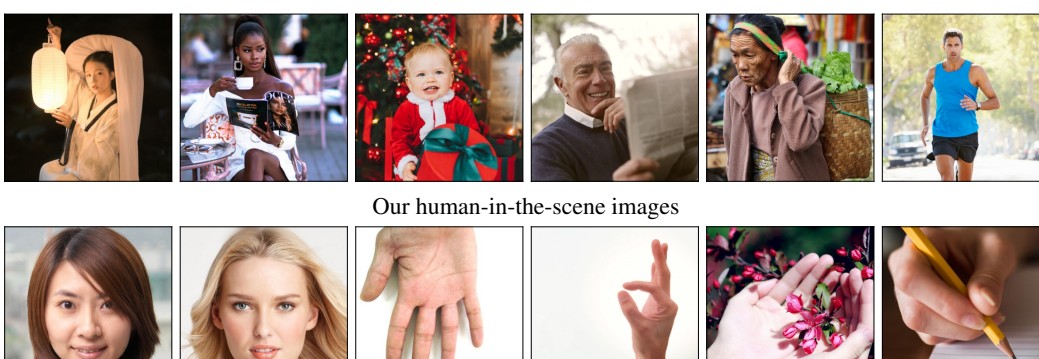

Our human-in-the-scene images

Close-up of face images          Close-up of hand images

Figure 3: Some showcases of our human-centric dataset.

In the end, we propose a simple yet effective method called **M**ixture **o**f **L**ow-rank **E**xperts (**MoLE**). Our key insight is to regard low-rank modules trained on customized datasets as specialized experts of MoE and adaptively activate them in a soft form. By adopting SD v1.5 as a showcase, our method basically contains three stages: We fine-tune SD v1.5 on collected dataset to complement sufficient human-centric prior; Then we use two close-up datasets, *i.e.*, close-ups of face and hand images, to train two low-rank experts separately; Finally, we formulate these two low-rank experts in an MoE form and integrate them with the base model in an adaptive soft assignment manner. To evaluate MoLE, we construct two human-centric benchmarks using prompts from COCO Caption [4] and DiffusionDB [48]. The results based on SD v1.5, v2.1, and XL consistently suggest the superiority and generalization of MoLE. Our contribution can be summarized as:

• We carefully collect a human-centric dataset comprising over one million high-quality images to provide sufficient human-centric prior. Importantly, we include two high-quality close-up of face and hand subsets, especially for hand. To the best of our knowledge, such a high-quality close-up hand dataset is absent in prior related studies. Click here to have a look.

• We find the low-rank refinement phenomenon and are inspired to propose MoLE, a simple yet effective method. MoLE integrates low-rank modules trained on customized hand and face datasets as experts in an MoE framework with soft assignment [4] to enable flexible activation.

• We construct two human-centric evaluation benchmarks from DiffusionDB and COCO Caption. MoLE consistently demonstrates improvement over the state-of-the-art and exhibits broad generalization across SD v1.5, v2.1, and XL in human-centric generation.

## 2 Related Work

**Text-to-image generation.** Diffusion model [17, 46], especially in text-to-image generation, has been widely used since its proposal, *e.g.*, GLIDE [34] with classifier-free guidance [18], Imagen [42] with a large T5 text encoder [37], Stable Diffusion [40] using VAE encoded latents, and DALL-E 2 [38] using CLIP [36]. Different from them, our work primarily focuses on human-centric text-to-image generation. Though a concurrent work [28] also aims to improve human-centric generation, it does not explicitly consider the issue of face and hand and thereby the related generation is still unnatural. In contrast, MoLE is specially designed for this issue.

**Mixture-of-Experts.** In MoE [20], different subsets of data or contexts may be better modeled by distinct experts. Theoretically, MoE could scale model capability with little cost by using sparsely-gated MoE layer [44]. Recently, MoE has been adapted in generation tasks [12, 2]. For example, ERNIE-ViLG [12] uniformly divides the denoising process into several distinct stages, with each being associated with a specific model. eDiff-i [2] calculates thresholds to separate the whole process into three stages. Differing from employing experts in divided stages, we consider low-rank modules trained by customized datasets as experts to adaptively refine generation. For more discussion about related work, we put in Appendix A.11.

---

[4]Considering that the human face and hand are generally the most frequently observed parts in an image and their bad cases are also extensively discussed or complained in image generation communities, thereby in this work, we primarily focus on the two most important and urgent parts. Our work could also easily involve other human parts, *e.g.*, feet, by collecting a close-up of feet dataset, training an extra low-rank feet expert, and accordingly modifying the parameter of the soft assignment.

|  | BLIP2 | ClipCap | MiniGPT-4 | LLaVA |
|---|---|---|---|---|

a man in a black jacket leaning against a wall

a man standing in front of a wall with graffiti on it and wearing a black jacket and black shirt

The image shows a young man standing in front of a graffiti-covered brick wall, wearing a black leather jacket and looking off into the distance with a serious expression on his face.

A young man with a black jacket and a black hoodie is standing in front of a graffiti-covered brick wall, looking at the camera with a serious expression.

a little girl is playing in the water

a little girl is playing in the water by a pond with a rock and grass in the background and a waterfall in the foreground

This image shows a small child sitting on a rock in a pond, looking into the water with a curious expression on their face. The child is wearing a brown dress and has dark hair. The pond is surrounded by trees and rocks, and there is a small waterfall in the background. The water in the pond is clear and reflective, creating a mirror effect. The child's reflection can be seen in the water. The overall mood of the image is peaceful and serene.

A young girl in a brown dress squatting near a small pond, looking into the water and admiring her reflection, while a fountain is also present in the background.

Prompt for MiniGPT-4 and LLaVA:
Describe this image in one sentence with details.

Figure 4: The results of four captioning models. Texts in red are inaccurate descriptions and texts in green are detailed correct descriptions. LLaVA presents a good balance between the level of detail and error rate, and thus is chosen for captioning our dataset.

## 3 Human-centric Dataset

**Overview.** Our human-centric dataset involves over one million high-quality images, containing three parts (See Sec 3.1). As shown in Fig 3, these images are diverse w.r.t. occasions, activities, gestures, ages, genders, and racial backgrounds. Specifically, approximately 57.33% individuals identify as White, 14.68% as Asian, 9.98% as Black, 5.11% as Indian, 5.52% as Latino Hispanic, and 7.38% as Middle Eastern [5]. Approximately 58.18% are male and 41.82% are female. For age, approximately 0.93% are babies (0-1 years old), 3.55% are kids (2-11 years old), 4.60% are teenagers (12-18 years old), 84.86% are adults (18-60 years old), and 6.06% are elderly (over 60 years old).
**Ethical & legal compliance.** Our collection is in compliance with the ethics and law as all images are collected from websites under Public Domain CC0 1.0 [6] license that allows free use, redistribution, and adaptation for non-commercial purposes. To avoid concerns, please see our license and privacy statement in Appendix A.2. Note that this dataset is allowed for academic purposes only. When using it, the users are requested to ensure compliance with ethical and legal regulations.

### 3.1 Human-centric Dataset Constitution

**Human-in-the-scene images.** We primarily collect high-resolution human-centric images from Internet and the image resolution is basically over $1024 \times 2048$, providing sufficient priors for diffusion models. To enable training, we use a sliding window ($1024 \times 1024$) to crop the image to maintain as much information as possible. However, for an image, high resolution does not mean high quality. Therefore, we train a VGG19 [45] to filter out blurred images. Additionally, considering the crop operation could generate images that are full of background or contain little information about people, we train a VGG19 [45] to filter out such bad cases [7]. To ensure the quality, we repeat the two processes multiple times until we do not find any case mentioned above in three times of random sampling. By employing these strategies, we can remove amounts of noise and useless images, thereby guaranteeing the image quality.

**Close-up of face images.** The face dataset contains two sources: the first is from Celeb-HQ [22] in which we choose images of high quality with size $1024 \times 1024$; The second is from Flickr-Faces-HQ (FFHQ) [23]. We sample images covering different skin color, age, sex, and race. There are around 6.4k face images. We do not sample more face images as it is sufficient for low-rank expert training.

**Close-up of hand images.** The hand dataset contains three sources: the first is from 11k Hands [1] where we randomly sample around 1k high-quality images and manually crop them to square; The second is from the Internet where we collect hand images of high quality and resolution with simple backgrounds and use YOLOv5 [9] to detect hands and crop them with details maintained; The third is from human-in-the-scene images (before processing) where we sample 8k images. We check every image and manually crop the hand of the image to square if the image is appropriate and the hand is clear. In this close-up hand dataset, there are abundant hand gestures and scenarios shown in Fig 3, *e.g.*, holding a flower, writing, *etc*. There are 7k high-quality hand images. To the best of our knowledge, such a high quality close-up hand dataset is absent in prior related studies.

---

[5]Similar to famous FFHQ dataset [23], our dataset inevitably inherits the potential bias of target websites.
[6]https://creativecommons.org/publicdomain/zero/1.0/
[7]See Appendix A.3 for training details of above two filters and illustration of bad samples.

## 3.2 Image Caption Generation

When collecting the dataset, we primarily consider image quality and resolution, neglecting whether it is text paired so as to increase image amount. Thus, producing a caption for each image is required. We investigate four recently proposed SOTA models including BLIP-2 [26], ClipCap [32], MiniGPT-4 [53], and LLaVA [29]. We show several cases in Fig 4. One can see that BLIP-2 usually produces simple descriptions and ignores details. ClipCap has a better performance but still lacks sufficient details along with the wrong description. MiniGPT4, although gives detailed descriptions, is inclined to spend a long time (17s on average) generating long and inaccurate captions that exceed the input limit (77 tokens) of the Stable Diffusion CLIP text encoder [36]. In contrast, LLaVA produces neat descriptions in one sentence with accurate details in a short period (3-5s). Afterward, we manually streamline long LLaVA caption with a new shorter caption by ourselves while aligning with the content of the image. We also remove unrelated and uninformative text patterns, *e.g.*, "The image features that . . . ", "showcasing . . . ", "creating . . . ", "demonstrating . . . ", *etc*. To further ensure the caption alignment of LLaVA, we use CLIP to filter image-text pairs with lower scores.

# 4 Method

## 4.1 Preliminary

**Low-rank Adaptation (LoRA).** Given a customized dataset, instead of training the entire model, LoRA [19] is designed to fine-tune the "residual" of the model, *i.e.*, $\triangle W$:

$$W^{'} = W + \triangle W \tag{1}$$

where $\triangle W$ is decomposed into low-rank matrices: $\triangle W = AB^T$ ($A \in \mathbb{R}^{n \times d}$, $B \in \mathbb{R}^{m \times d}$, $d < n$, and $d < m$). During training, we can simply fine-tune $A$ and $B$ instead of $W$, making fine-tuning on customized dataset memory-efficient. In the end, we get a small model as $A$ and $B$ are much smaller than $W$.

**Mixture-of-Experts (MoE).** MoE [20, 44, 24] is designed to enhance the predictive power of models by combining the expertise of multiple specialized models. Usually, a central "gating" model $G(.)$ selects which specialized model to use for a given input:

$$y = \sum_{i=1} G(x)_i E_i(x) \,. \tag{2}$$

When $G(x)_i = 0$, the corresponding expert $E_i$ will not be activated.

## 4.2 Mixture of Low-rank Experts

Motivated by the two potential factors discussed in Sec 1, our method contains three stages as shown in Fig 5. We describe each stage below and put the training details in Appendix A.1.

***Stage 1: Fine-tuning on Human-centric Dataset.*** The overall poor performance of human-centric generation could be attributed to the absence of large-scale high-quality datasets. Considering such a pressing need, our work bridges this gap by providing a carefully collected dataset that contains around one million human-centric images of high quality. To learn as much prior as possible, we adopt SD v1.5 as a baseline and leverage the whole human-centric datasets to fine-tune. Concretely, we fine-tuning the UNet modules [41] (and text encoder) while fixing the rest parameters. The well-trained model is then sent to the next stage.

***Stage 2: Low-rank Expert Generation.*** To construct MoE, in this stage, our goal is to prepare two experts that are supposed to contain abundant knowledge about the corresponding part. To achieve this, we train two low-rank modules using two customized datasets. One is the close-up face dataset. The other is the close-up hand dataset that contains abundant hand gestures, full details with simple backgrounds, and interactions with other objects. We then use the two datasets to train two low-rank experts with SD v1.5 trained in stage 1 as the base model. The low-rank experts are expected to focus on the generation of face and hand and learn useful context.

***Stage 3: Soft Mixture Assignment.*** This stage is motivated by the low-rank refinement phenomenon in Fig 2 where a specialized low-rank module using a proper scale weight is able to refine the corresponding part of a person. Hence, the key is to activate different low-rank modules with suitable weights. From this view, MoE naturally stands out and we novelly regard a low-rank module trained

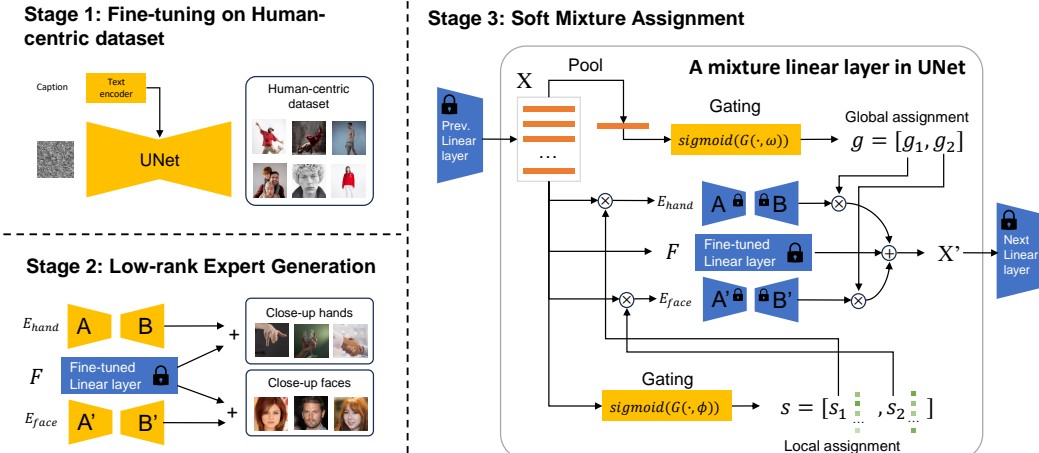

Figure 5: The framework of MoLE. $X$ is the input of any linear layers in UNet. $A$ and $B$ are low-rank matrices.

on a customized dataset, *e.g.*, face dataset, as an expert and formulate in a MoE form. Moreover, for a person, face and hand would appear in an image simultaneously while hard assignment mode in MoE only allows one expert accessible to the given input. Hence, inspired by Soft MoE [35], we adopt a soft assignment, allowing multiple experts to handle input simultaneously. Further, considering that the face and hand would be a part of the whole image (local) or occupy the whole image (global), besides global assignment in original MoE, we novelly introduce local assignment. Specifically, considering a linear layer $F$ from UNet and its input $X \in \mathbb{R}^{n \times d}$ where $n$ is the number of token and $d$ is the feature dimension, we illustrate local and global assignment respectively.

**For local assignment**, we employ a local gating network that contains a learnable gating layer $G(\ , \phi)$ ($\phi \in \mathbb{R}^{d \times e}$, $e$ is the number of experts and here $e$ is 2, below is the same.) and a sigmoid function. The gating network is to produce two normalized score maps $s = [s_1, s_2]$ ($s \in \mathbb{R}^{n \times e}$, here $e$ is 2) for two low-rank experts as formulated:

$$s = \text{sigmoid}(G(X\ , \phi) \tag{3}$$

**For global assignment**, we use a global gating network including an AdaptiveAvePool, a learnable gating layer $G(\ , \omega)$ ($\omega \in \mathbb{R}^{d \times e}$, here $e$ is 2), and a sigmoid function. This gating network is to produce two global scalars $g = [g_1, \ g_2]$ ($g \in \mathbb{R}^e$, $e$ is 2) for two experts as formulated:

$$g = \text{sigmoid}(G(\text{Pool}(X)\ , \omega)) . \tag{4}$$

*The soft mechanism is built on the fact that each token can adaptively determine how much (weight) should be sent to each expert by the* sigmoid *function.* And intuitively, the weight of every token for two experts is independent as face and hand experts are not competitors during generation. Thus we do not use softmax.

**For combination**, we first send $X$ to each low-rank expert $E_{\text{face}}$ and $E_{\text{hand}}$ respectively, use $s_1$ and $s_2$ ($\mathbb{R}^{n \times 1}$) to perform element-wise multiplication (local assignment), and also perform global control by scalars $g_1$ and $g_2$ (global assignment) [8]:

$$Y_1 = E_{\text{face}}(X \cdot s_1 \cdot g_1) = g_1 \cdot E_{\text{face}}(X \cdot s_1) \tag{5}$$
$$Y_2 = E_{\text{hand}}(X \cdot s_2 \cdot g_2) = g_2 \cdot E_{\text{hand}}(X \cdot s_2) \tag{6}$$

Then we add $Y_1$ and $Y_2$ back to the output of a linear layer $F$ from UNet with $X$ as input, producing a new output $X'$:

$$X' = F(X) + Y_1 + Y_2 \tag{7}$$

In the end, to endow the model with the capability to adaptively activate experts, we use our human-centric dataset to train learnable gating layers while freezing the base model and two low-rank experts.

## 5 Experiment

### 5.1 Evaluation Benchmarks and Metrics

Considering that our work primarily focuses on human-centric image generation, before presenting our experiment, we introduce two customized evaluation benchmarks. Additionally, since our

---

[8]Recalling that each expert is two low-rank matrices, $g_1$ and $g_2$ can transition from within $E_{\text{face}}$ and $E_{\text{hand}}$ to outside of them.

Table 1: The performance of MoLE on COCO Human Prompts and DiffusionDB Human Prompts.

| Model | COCO Human Prompts | |
| --- | --- | --- |
| | HPS (%) | IR (%) |
| VQ-Diffusion | $19.21 \pm 0.04$ | $-12.51 \pm 2.44$ |
| Versatile Diffusion | $19.75 \pm 0.09$ | $-8.81 \pm 1.40$ |
| SDXL | $20.84 \pm 0.06$ | $73.34 \pm 2.29$ |
| SD v1.5 | $19.91 \pm 0.09$ | $28.34 \pm 1.40$ |
| MoLE (SD v1.5) | $20.27 \pm 0.07$ | $33.75 \pm 1.49$ |
| MoLE (SDXL) | $21.36 \pm 0.02$ | $98.52 \pm 0.61$ |

| Model | DiffusionDB Human Prompts | |
| --- | --- | --- |
| | HPS (%) | IR (%) |
| VQ-Diffusion | $19.00 \pm 0.02$ | $-18.42 \pm 1.49$ |
| Versatile Diffusion | $20.09 \pm 0.04$ | $-29.05 \pm 2.72$ |
| SDXL | $21.51 \pm 0.07$ | $87.88 \pm 2.53$ |
| SD v1.5 | $20.29 \pm 0.01$ | $-2.72 \pm 1.66$ |
| MoLE (SD v1.5) | $20.62 \pm 0.04$ | $4.36 \pm 1.36$ |
| MoLE (SDXL) | $22.35 \pm 0.01$ | $105.25 \pm 1.15$ |

Figure 6: User study in four aspects.

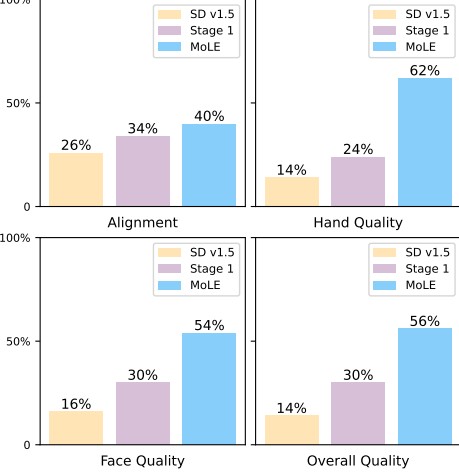

generated images are human-centric, they should intuitively meet human preferences. Hence, we primarily adopt two human preference metrics including Human Preference Score (HPS) [49] and ImageReward (IR) [50]. We describe all of them below. Besides the two metrics, we also perform user studies by inviting people to compare the generated images with their own preferences.

**Benchmark 1: COCO Human Prompts.** We construct this benchmark by leveraging the caption in COCO Caption [4] that has been widely used in previous work [49, 50, 6]. Concretely, we use the captions in the COCO val set, and preserve the caption that contains human-related words, *e.g.*, woman, man, kid, girl, boy, person, teenager, *etc*. In the end, we have around 60k prompts left, dubbed as COCO Human Prompts.

**Benchmark 2: DiffusionDB Human Prompts.** DiffusionDB [48] is the first real-users-specified large-scale text-to-image prompt dataset containing around 14 million prompts. We construct this benchmark by leveraging its 2M caption set version. Concretely, we first filter out the NSFW prompts by the indicator provided in DiffusionDB. Then we preserve captions containing human-related words. We filter out prompts containing special symbol, *e.g.*, [, ], *etc*. In the end, we have around 64k prompts left, dubbed as DiffusionDB Human Prompts.

**Metric 1: Human Preference Score (HPS).** Human Preference Score (HPS) [49] measures how images present with human preference. It leverages a human preference classifier fine-tuned on CLIP [36].

**Metric 2: ImageReward (IR).** Different from HPS, ImageReward (IR) [50] is built on BLIP [27]. IR is a zero-shot evaluation metric for understanding human preference in text-to-image synthesis.

## 5.2 Main Results

**Superiority.** To evaluate the performance, following previous work [40, 50, 49, 6] we randomly sample 3k prompts from COCO Human Prompts benchmark and 3k prompts from DiffusionDB Human Prompts benchmark to generate images and calculate metrics, HPS and IR, and compare MoLE with open-resource SOTA method with different model structures including VQ-Diffusion [13], Versatile Diffusion [51], our baseline SD v1.5 [40] and its largest variant SDXL. We repeat the process three times and report the averaged results and standard error in Tab 1. MoLE outperforms VQ-Diffusion and Versatile Diffusion and significantly improves our baseline SD v1.5 in both metrics, implying that MoLE could generate images that are more natural to meet human preference. In Appendix A.5, we also evaluate on CLIP-T [25], FID [16], and Aesthetic Score [43], to present a comprehensive comparison. Besides, we illustrate generated images of different models in Fig 1 and Fig 13. Though MoLE is inferior to SDXL in HPS and IR [9], we qualitatively compare the face and hand of generated images from MoLE and SDXL in Fig 1 and Fig 13, and our results look more realistic with natural hand/face even under high HPS gap (in 2nd row Fig 1 20.39 *vs*. 22.38). Even though, to provide a more convincing demonstration of the efficacy of our method, we proceed to implement MoLE on SDXL (See Generalization part below). We also compare MoLE with relevant methods like HanDiffuser [33] and HyperHuman [30] in Appendix A.5. Finally, MoLE is resource-friendly and can be trained in a single A100 80G GPU.

---

[9]MoLE is built on SD v1.5, which has a significant gap in model size (5.1G *vs*. 26.4G) and output resolution (512 *vs*. 1024) compared to SDXL.

**Generalization.** Besides SD v1.5, to verify the generalization of our method, we further construct MoLE on SDXL in Tab 1. We also consider SD v2.1 and transformer-based PixArt-$\alpha$ in Appendix A.5. In Tab 1, we see that based on SDXL, MoLE outperforms SDXL by a large margin. Similar conclusion can be observed in SD v2.1 and PixArt-$\alpha$. These results demonstrate great generalization of our method. Also, we qualitatively compare images generated by MoLE (SDXL) and SDXL in Appendix A.8 where MoLE generates more natural hands/faces.

## 5.3 Ablation Study

**Stage Enhancement.** To figure out how each stage of MoLE enhances the generation performance, we use MoLE built on SD v1.5 to conduct the experiments on COCO Human Prompts by randomly sampling 3k prompts to generate images from each stage with the same seed in Sec 5.2 and calculate the HPS and IR. The whole process repeats three times as well. The results are reported in Tab 2. In Fig 7, we also illustrate examples generated by different stages with the same seeds to intuitively show the role of each stage. We see that fine-tuning on human-centric dataset (Stage 1) is effective in improving overall quality, proved by enhanced HPS and IR, implying the importance of the dataset. However, when adding Stage 2, *i.e.*, both experts are employed, the performance drops. We speculate that, due to training on close-up datasets, the employed experts tend to mimic their training data's distribution and thereby harm the generation process, *e.g.*, in Fig 7 Stage 2, close-up image of face resemble the distribution of face image in FFHQ dataset [23]. Luckily, as shown in Fig 7 MoLE, adding Stage 3 allows model to adaptively activate two experts with soft assignments and thereby alleviates this issue, verifying its importance. Also, Stage 3 refines human face and hand compared to Stage 1 and thereby outperforms Stage 1 on HPS and IR in Tab 2. Finally, to further verify MoLE's advantage over SD v1.5 and demonstrate the significance of Stage 3, we conduct user studies by sampling 20 sets of images from SD v1.5, fine-tuned SD (Stage 1), and MoLE, and inviting 50 participants to select the best one from each set according to their preference in four aspects respectively. In Fig 6 we see that Stage 1 improves overall performance but is limited in improving face and hand quality while MoLE obtains highest voting, especially in hand and face quality, indicating that Stage 1 alone is insufficient, and Stage 3 also holds significance.

Table 2: Ablation study on each stage using COCO Human Prompts.

| Stage | HPS (%) | IR (%) |
|---|---|---|
| SD v1.5 | $19.91 \pm 0.09$ | $28.34 \pm 1.40$ |
| +Stage 1 | $20.16 \pm 0.09$ | $31.01 \pm 1.75$ |
| +Stage 2 | $19.94 \pm 0.07$ | $25.66 \pm 2.72$ |
| +Stage 3 | $20.27 \pm 0.07$ | $33.75 \pm 1.49$ |

Table 3: Ablation study on assignment manner using COCO Human Prompts.

| Method | HPS (%) | IR (%) |
|---|---|---|
| SD v1.5 | $19.91 \pm 0.09$ | $28.34 \pm 1.40$ |
| Local | $20.19 \pm 0.04$ | $31.98 \pm 2.41$ |
| Global | $20.20 \pm 0.02$ | $32.20 \pm 0.86$ |
| Both | $20.27 \pm 0.07$ | $33.75 \pm 1.49$ |

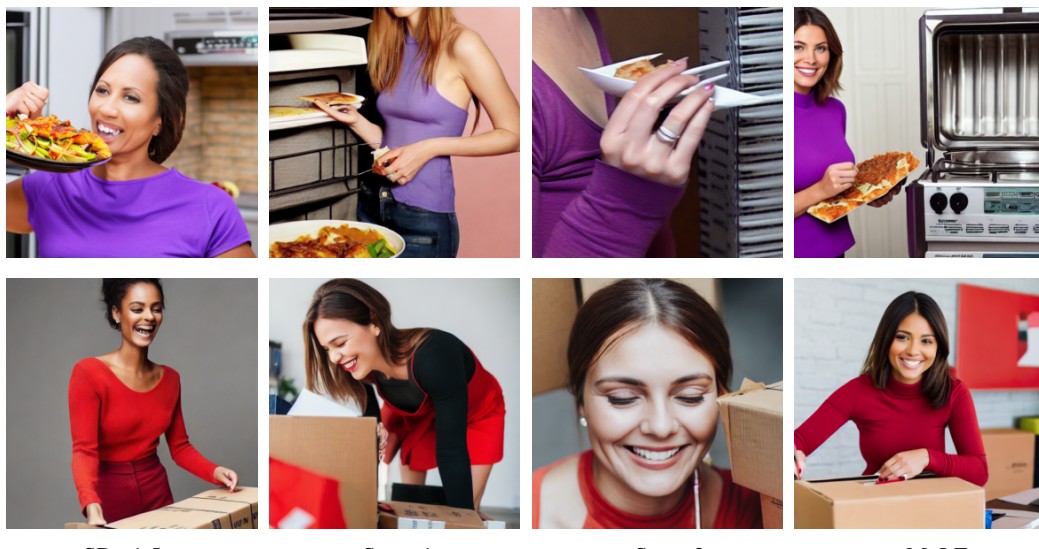

| SD v1.5 | Stage 1 | Stage 2 | MoLE |

Figure 7: Showcases of image generated by different stages. Top prompt: a woman in a purple top pulling food out of a oven. Bottom prompt: smiling woman in red top putting items in a box.

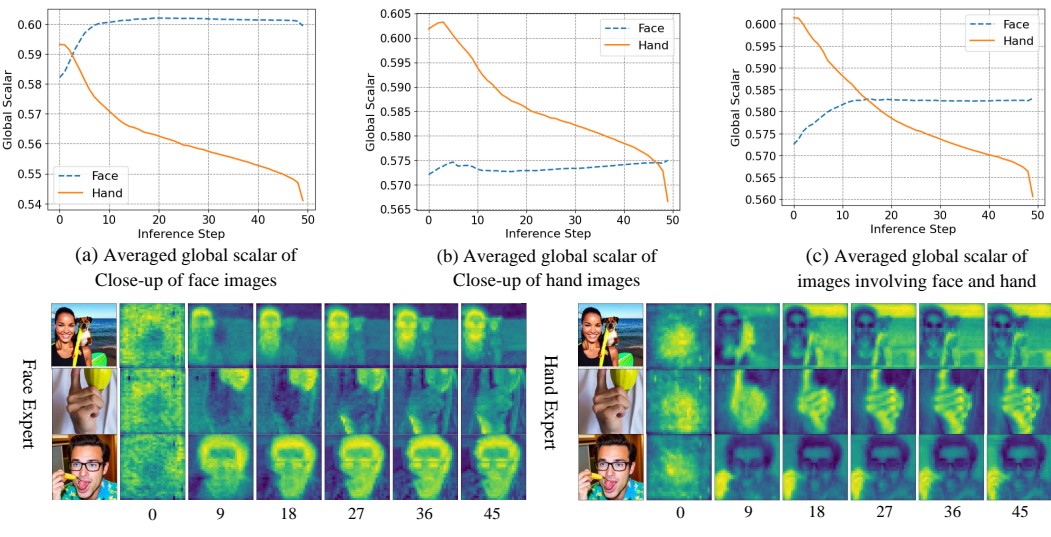

(a) Averaged global scalar of Close-up of face images

(b) Averaged global scalar of Close-up of hand images

(c) Averaged global scalar of images involving face and hand

(d) Visualization of score map from face expert (left) and hand expert (right) in different inference step.

Figure 8: The averaged global and local assignment weights in different inference steps.

**Mixture Assignment.** In MoLE, we use two kinds of mixture manners including local and global assignment. Hence, we ablate the two assignments and present the results in Tab 3. It can be seen that both local and global assignments can enhance performance. When combining them together, the performance is further improved, indicating the effectiveness of our mixture manners. Moreover, we illustrate how the two assignments work in Fig 8. For global assignment, we average the global scalars of 20 close-up face images, 20 close-up hand images, and 20 normal human images involving hand and face respectively in every inference step in Fig 8 (a), (b), and (c). In (a) and (b), when generating different close-ups, the corresponding expert generally produces higher global value, implying that global assignment is content-aware. In (c), $E_{\text{face}}$ and $E_{\text{hand}}$ achieve a balance. Besides, as inference progresses the global scalar of $E_{\text{hand}}$ always drops while that of $E_{\text{face}}$ is relatively flat. We speculate, in light of the diversity of hands (*e.g.*, various gestures), $E_{\text{hand}}$ tends to establish general content in the early stage while $E_{\text{face}}$ must meticulously fulfill facial details throughout the denoise process due to fidelity requirement. For local assignment, we visualize averaged score maps of sampled images from the two experts respectively in Fig 8 (d). We see that as inference progresses, local assignment of the two experts can highlight and gradually refine corresponding parts, verifying its effectiveness. We also provide the distribution of the local weight sent to each expert in Appendix A.6. Additionally, to understand the importance of using two experts on the model's performance, we train only one expert using all close-up images and put the comparison results in Appendix A.7.

## 5.4 More Visualizations and Analysis

We show more human-centric images with natural face and hand in Appendix A.8. Surprisingly, MoLE can also generate non-human-centric images [10] in Appendix A.9, *e.g.*, animals and scenery. In the end, we analyze failure cases in Appendix A.10, which we find are attributed to the large L2 norm of outputs from face and hand experts.

## 6 Discussion

To further highlight our method contribution, below we present a comprehensive discussion on distinctions between MoLE and conventional MoE methods from three aspects. We also discuss the contribution of our curated human-centric dataset and analysis about ratios of different races in Appendix A.12. Firstly, from the aspect of training, MoLE independently trains two experts to learn completely different knowledge using two customized close-up datasets. In contrast, conventional MoE methods simultaneously train experts and base models using the same dataset. Secondly, from the aspect of expert structure and assignment manner, MoLE simply uses two low-rank matrices while conventional MoE methods use MLP or convolutional layers. Moreover, MoLE combines local

---

[10]We speculate this is because the human-centric dataset also contains these entities that interact with humans in an image. Hence, the model learns these concepts. However, it is worth noting that MoLE may not be better at generic image generation than the generic models as MoLE is trained on a human-centric dataset.

and global assignments together for a finer-grained assignment while conventional MoE methods only use global assignment. Finally, from the aspect of applications in computer vision, MoLE is proposed for text-to-image generation while conventional MoE methods are mainly used in object recognition, scene understanding, *e.g.*, V-MoE [39]. Though MoE recently has been employed in image generation, *e.g.*, ERNIE-ViLG [12] and eDiff-i [2] that employ experts in divided stages, MoLE differs from them – inspired by low-rank refinement in Fig 2, MoLE consider low-rank modules trained by customized datasets as experts to adaptively refine image generation.

## 7    Conclusion

In this work, we primarily focus on the human-centric text-to-image generation that has important real-world applications but often suffers from producing unnatural results due to insufficient prior, especially the face and hand. To mitigate this issue, we carefully collect and process one million high-quality human-centric images, aiming to provide sufficient prior. Besides, we observe that a low-rank module trained on a customized dataset, *e.g.*, face, has the capability to refine the corresponding part. Inspired by it, we propose a simple yet effective method called Mixture of Low-rank Experts (MoLE) that effectively allows diffusion models to adaptively select experts to enhance the generation quality of corresponding parts. We also construct two customized human-centric benchmarks from COCO Caption and DiffusionDB to verify the superiority of MoLE.

## 8    Limitation & Future Work

Honestly, although our experiments confirm MoLE's effectiveness in enhancing human-centric image generation, there is still considerable room for improvement. Our method struggles with scenarios involving multiple individuals, likely due to our dataset being primarily single-person images and uncertainty about the applicability of observations in Fig 2 to such cases. Additionally, in generating images using identical prompts, we observe that only about 25% of MoLE's results are of high quality, a remarkable improvement over SDXL's 10%, but still below practical standards. Potential reasons include insufficient close-up data for hands and faces and the need for further tuning of hyperparameters. Future work will focus on model optimization, improving data quality, and enhancing dataset diversity to better represent various demographics and real-world scenarios.

## 9    Broader Impact

MoLE mainly focuses on enhancing human-centric text-to-image generation in diffusion models. It refrains from introducing any harmful content to the community and society. However, though MoLE may not introduce more biases on race, it also inherits the biases in the training data like pervious methods. Hence it will be more meaningful to enhance the diversity of our collected dataset to represent different demographics and real-world scenarios better. As for other impacts such as fake faces, it also inevitably generates fake faces like other generative models, which requires users to leverage these generated images carefully and legally. We highlight that these issues also warrant further research and consideration. We maintain transparency in our methods with open-source code and dataset composition, allowing for continuous improvement based on community feedback.

**Acknowledgments**

The authors thank the anonymous reviewers for constructive comments. The authors also thank Qi Zhang, Xin Li, Boqiang Duan, Teng Xi, and Gang Zhang for discussion and help.

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

# A Appendix

## A.1 Implementation Details

**Stage 1: Fine-tuning on human-centric Dataset.** We use Stable Diffusion v1.5 as base model and fine-tune the UNet (and text encoder) with a constant learning rate $2e - 6$. We set batch size to 64 and train with the Min-SNR weighting strategy [14]. The clip skip is 1 and we train the model for 300k steps using Lion optimizer [5]. We use this stage on SD v1.5. For SD v2.1 and SDXL, we do not use this stage to fine-tune the base models as their overall human-centric generation is relatively satisfying but only looks poor on the details of face and hand.

**Stage 2: Low-rank Expert Generation.** For face expert, we set batch size to 64 and train it 30k steps with a constant learning rate $2e - 5$. The rank is set to 256 and AdamW optimizer is used. For hand expert, we set batch size to 64. Since hand is more sophisticated than face to generate, we train it 60k steps with a smaller learning rate $1e - 5$. The rank is also set to 256 and AdamW optimizer is used. For both experts, we only add low-rank module to UNet. And the two experts are both built on the fine-tuned base model in Stage 1.

**Stage 3: Mixture Adaptation.** In this stage, we use the batch size 64 and employ AdamW optimizer. We use a constant learning rate $1e - 5$ and train for 50k steps.

## A.2 License and Privacy Statement

The human-centric dataset is collected from websites including seeprettyface.com, unsplash.com, gratisography.com, morguefile.com, pexels.com, *etc.* We use web crawler to download images only when it is allowed. Most images in these websites are published by their respective authors under Public Domain CC0 1.0 [11] license that allows free use, redistribution, and adaptation for non-commercial purposes. Seeprettyface.com require giving appropriate credit to the author by adding the sentence (# Thanks to dataset provider:Copyright(c) 2018, seeprettyface.com, BUPT_GWY contributes the dataset.) to the open-source code when using the images. When collecting and filtering the data, we are careful to only include images that, to the best of our knowledge, are intended for free use and redistribution by their respective authors. That said, we are committed to protecting the privacy of individuals who do not wish their images to be included. Besides, for images fetched from other datasets, *e.g.*, Flickr-Faces-HQ (FFHQ) [23], Celeb-HQ [22], and 11k Hands [1], we strictly follow their licenses and privacy. Note that this dataset is allowed for academic purposes only. When using it, the users are requested to ensure compliance with ethical and legal regulations. For the application for the usage of generated images and the dataset, we will carefully review the applicant's qualifications, purpose of use, possible risks [56, 54, 55], *etc.* Finally, we only allow authorized personnel to interact with the data.

## A.3 Filter Training and Illustrations of Negative Samples

In both case, to train the VGG19, we manually collect around 300 positive samples and 300 negative samples as training set, and we also collect around 100 positive samples and 100 negative samples as val set. When training the VGG19, we set the batch size to 128, set the learning rate to 0.001, and use random flip as the data augmentation method. We train the model for 200 epochs and use the best-performing model for subsequent classification. In Fig 9, we present the illustrations of negative samples during refining human-in-the scene subset.

## A.4 Comparison with Existing Related Datasets

We give a comparison of the differences between existing datasets like CosmicMan [28] and our newly collected dataset, which primarily lie in four aspects:

• From the aspect of image diversity, due to different motivations, CosmicMan only contains human-in-the-scene images while our dataset also involves two close-up datasets for face and hand, respectively. Moreover, to the best of our knowledge, the high-quality close-up hand dataset is absent in prior related studies.

---

[11] https://creativecommons.org/publicdomain/zero/1.0/

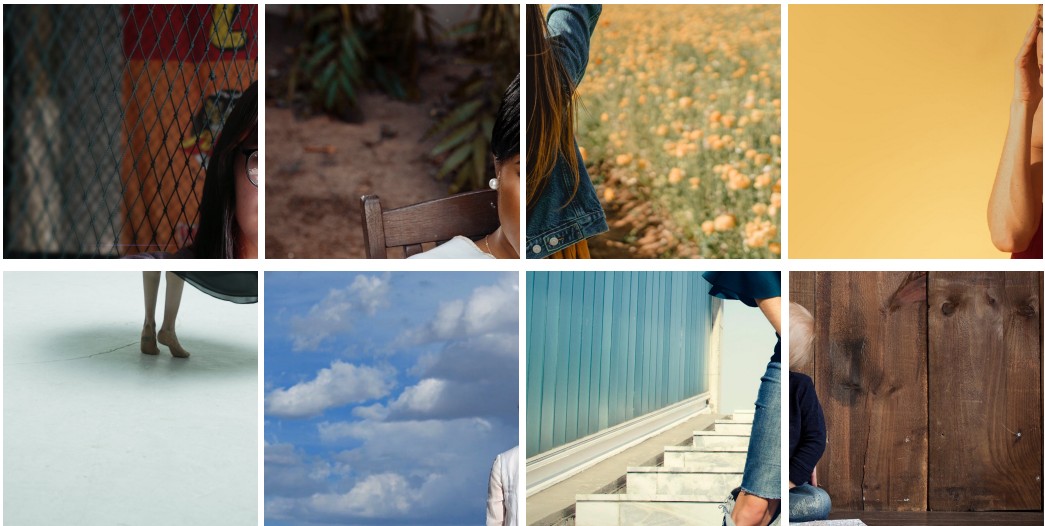

Figure 9: Illustrations of negative samples. Zoom in for a better view.

• From the aspect of image content distribution, there is a relatively severe gender imbalance in CosmicMan where female makes up a large proportion around 75% (See Fig 3 of Appendix in its paper) while our dataset is relatively balanced (58% vs 42%).

• From the aspect of image size, though CosmicMan and our dataset are both of high quality, our collected images (basically over 1024 × 2048) are relatively larger than CosmicMan whose average size is 1488 × 1255.

• From the aspect of data sources, our dataset is legally collected from various websites including unsplash.com, gratisography.com, morguefile.com, and pexels.com, *etc*., while CosmicMan is sourced from LAION-5B (See https://huggingface.co/datasets/cosmicman/CosmicManHQ-1.0). What sets our dataset apart is not just its wide collection, but also the freshness of the data. As a trade-off, the quantity of our dataset (1M) is relatively smaller than that of CosmicMan (5M).

### A.5 More Quantitative Comparisons

We perform four kinds of comparisons. We first evaluate MoLE and SD v1.5 on CLIP-T [25], FID [16], and Aesthetic Score [43] to present a comprehensive comparison. Specifically, We follow [25] and [21] using COCO Human Prompts and Human-centric datasets to evaluate the overall quality of generated images on CLIP-T and FID. We also use an aesthetic predictor [12] to generate the Aesthetic Score. All the results are reported in Tab 4. We find MoLE is also superior in CLIP-T and FID. We also find that MoLE is slightly inferior to SD v1.5 in aesthetic score. We deem that it is reasonable as SD v1.5 is especially fine-tuned on laion-aesthetics v2 5+ dataset in which each image's aesthetic score is evaluated with high aesthetics score by exactly the aesthetic predictor we used in this comparison.

Table 4: Comparisons between MoLE and SD v1.5 on CLIP-T, FID, and Aesthetic Score.

| Method | CLIP-T | FID | Aesthetic Score |
|---|---|---|---|
| SD v1.5 | 26.87 | 69.82 | 5.19 |
| MoLE | 27.33 | 64.37 | 5.18 |

Then, we construct MoLE based on SD v2.1 and evaluate MoLE and SD v2.1 using COCO Human prompt to further show the effectiveness and generalization of our method. The experiment shows that MoLE produces $20.45 \pm 0.09$ for HPS, outperforming SD v2.1 ($20.17 \pm 0.08$). For IR, MoLE produces $62.08 \pm 0.86$, outperforming SD v2.1 ($58.36 \pm 0.51$). These results further verifies the effectiveness of our method.

Besides, to verify the generalization of our method on transformer-based models, we also build our MoLE based on PixArt-XL-2-512×512 [3]. To compare the performance, we randomly sample 3k prompts from COCO Human Prompts and calculate HPS for MoLE (PixArt) and PixArt. The evaluation process is repeated three times. Our method achieves $21.79 \pm 0.03$ HPS (%) and outperforms PixArt ($21.33 \pm 0.08$ HPS). The result demonstrates the generalization of our method.

---

[12]https://laion.ai/blog/laion-aesthetics/

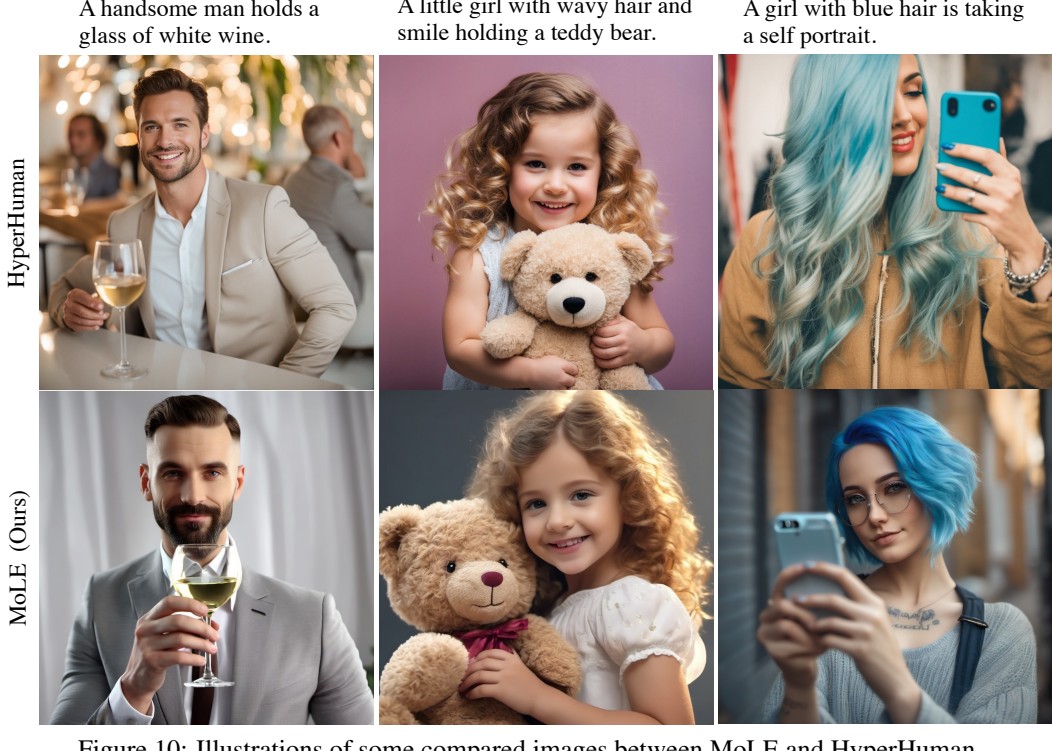

Figure 10: Illustrations of some compared images between MoLE and HyperHuman.

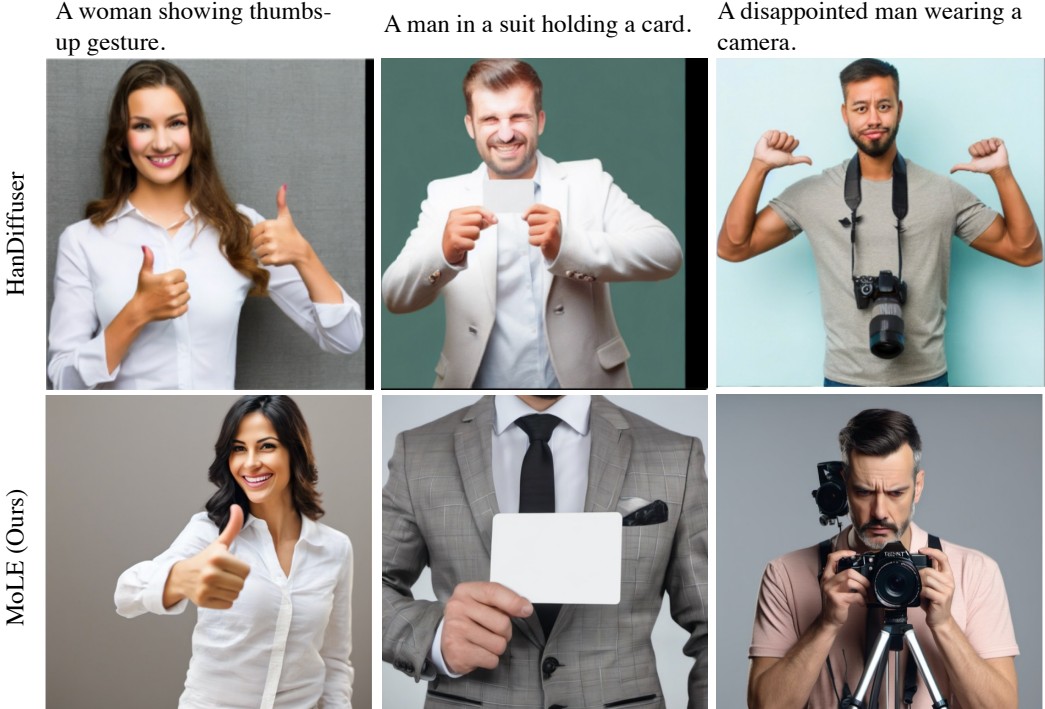

Figure 11: Illustrations of some compared images between MoLE and HanDiffuser.

Finally, we compare MoLE with state-of-the-art methods including HanDiffuser [33] and HyperHuman [30] through user study as their code has not been made available. Specifically, we invite 50 participants to compare the visualization presented in the two methods' papers with our generated images, respectively. In the user study, we prepare 10 MoLE-HyperHuman pairs and ask participants to select the better one from each pair according to their preference in terms of hand quality. Some compared images are presented in Fig 10 and Fig 11 to show the differences between our generated images and theirs. The results show that 58% of participants think our generated images are better than that of HyperHuman. Similarly, for HanDiffuser, we also prepare 10 MoLE-HanDiffuser

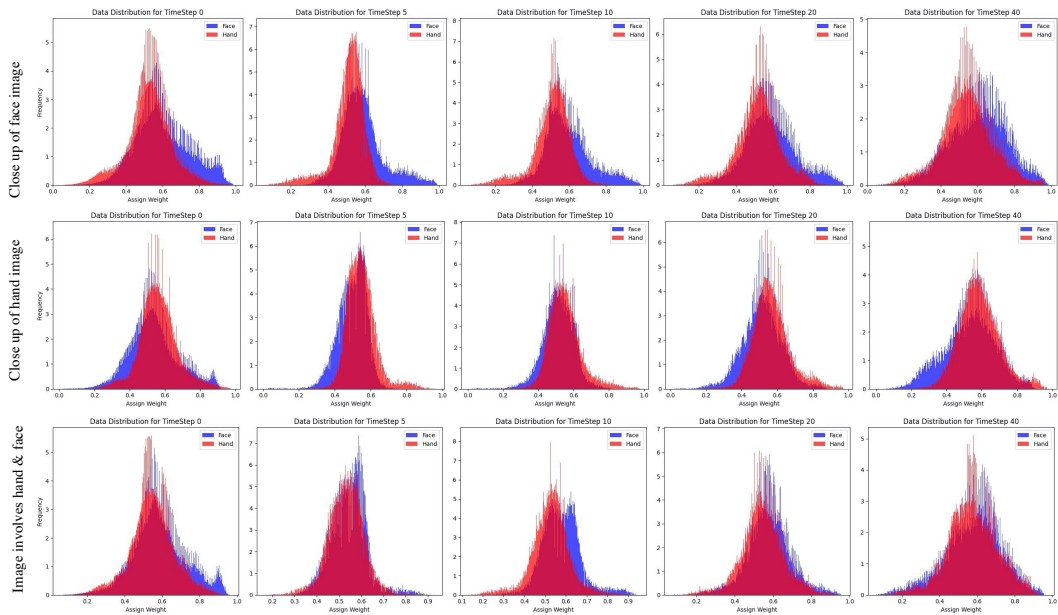

Figure 12: The distribution of the local weight sent to each expert in MoLE.

pairs and ask participants to select the better one. We find that 48% of participants vote for MoLE, slightly inferior to HanDiffuser (52%). All these results demonstrate that our method is effective and competitive with the state-of-the-art methods. More importantly, our method is user-friendly because both HanDiffuser and HyperHuman rely on additional conditions to enhance human and hand generation: HyperHuman takes text and skeleton as input; HanDiffuser needs text, a SMPL-H model, camera parameters, and hand skeleton. In contrast, MoLE only relies on text without the need for any additional conditions, offering greater flexibility and ease of use while maintaining competitive performance.

### A.6 Weight Distribution of Local Assignment

We provide the distribution of the local weight sent to each expert in Fig 12. To obtain this, we generate 10 samples for close-up images and normal human images, respectively, and collect local weights for each expert. In Fig 12, one can see that for close-up images, *e.g.*, face, the corresponding expert receives more weights of high value. We think this effectively demonstrates the efficacy of the soft assignment mechanism in MoLE, which adaptively activates the relevant expert to contribute more to the generation of close-up images. When generating normal human images involving face and hand, the two experts contribute equally, and generally, the face expert receives relatively more weights of high value as the area of face is typically larger than that of hand.

### A.7 Ablation Study on Experts

To understand the importance of using two experts on the model's performance, we train only one expert using all close-up images and compare the performance with that of two experts. We find that one expert achieves $20.19 \pm 0.03$ HPS(%), inferior to that of two experts ($20.27 \pm 0.07$ HPS), which demonstrates the necessity of using one expert for face and hand, respectively.

### A.8 More Visualization

We present more generated images and compare with other diffusion models in Fig 13. We provide more full-body images in Fig 14. Additionally, we illustrate more images generated by MoLE in Fig 17, Fig 18, Fig 19, and Fig 20. We also compare MoLE (build on SDXL) with SDXL in Fig 21, Fig 22, and Fig 23 where MoLE further enhances SDXL by generating more natural face/hand.

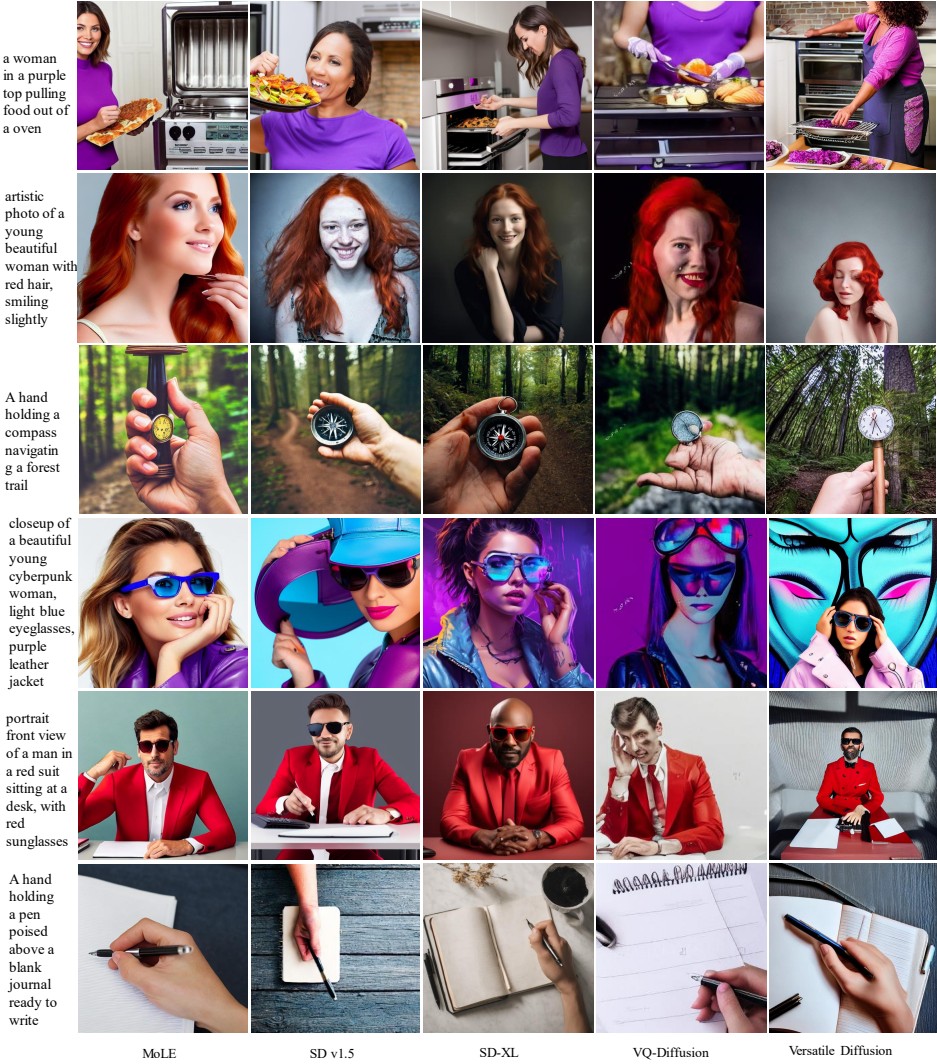

Figure 13: Comparison with other diffusion models. Zoom in for a better view.

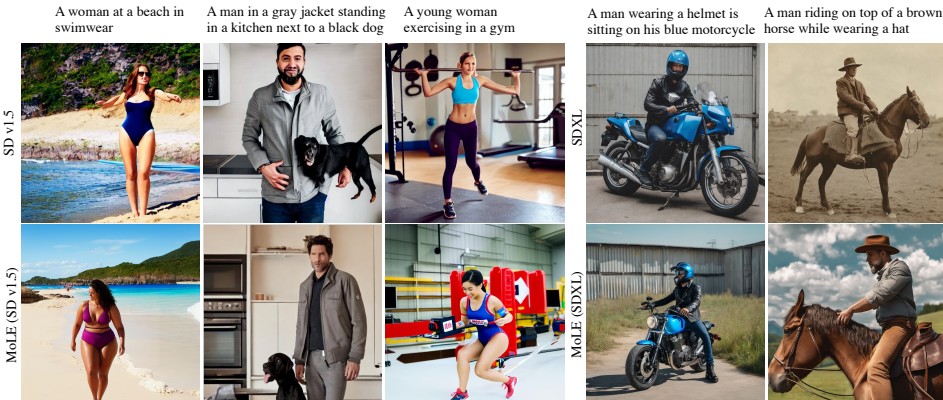

Figure 14: Comparisons about full-body images. Zoom in for a better view.

## A.9 Generic Image Generation

As shown in Fig 15, MoLE (fine-tuned SD v1.5) can also generate non-human-centric images, *e.g.*, animals, scenery, *etc*. The main reason could be that the human-centric dataset also contains these entities that interact with humans in an image. As a result, the generative model learns these concepts. However, intuitively, MoLE may not be better at generic image generation than the generic generative models as MoLE is trained on a human-centric dataset.

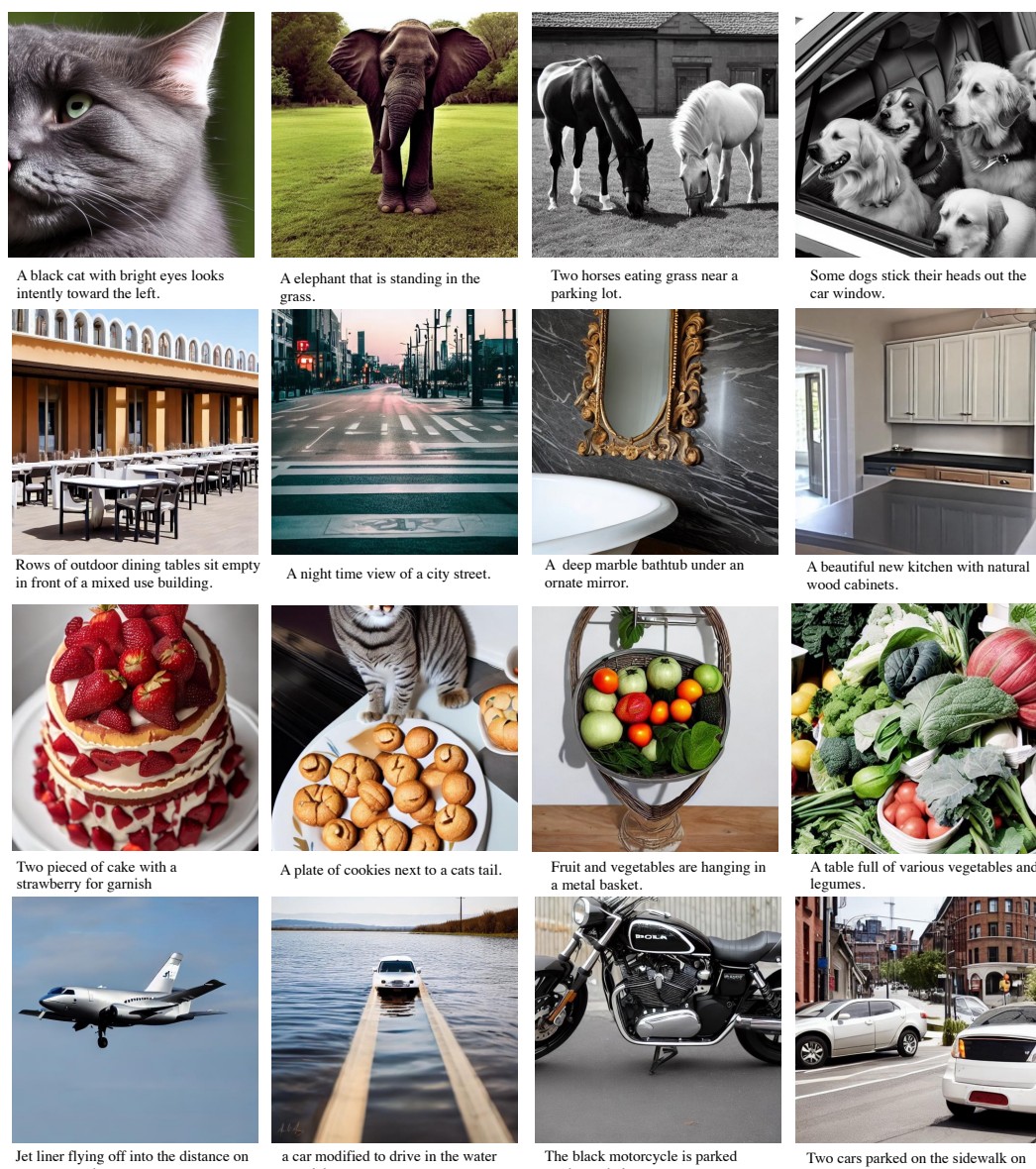

Figure 15: Generic image generation. Zoom in for a better view.

## A.10 Failure Case Analysis

We observe several failure cases involving unrealistic content. For example, we find a generated unrealistic images with a half-man in Fig 16 (a) whose prompt is "a young man skateboarding while listening to music". To figure out the reason, we sample 10 bad and normal cases, calculate their L2 norm of outputs from face and hand expert respectively, and visualize the averaged L2 norm across timestep as shown in Fig 16 (b). One can see that the bad cases generally have larger L2 norm for both experts, which indicates that the output from linear layer in Eq 7 is strongly influenced by the two experts. As a result, the generated images may be uncoordinated. We leave this as feature work.

## A.11 More Related Work

**Text-to-image generation.** Diffusion model [17, 46] has been widely used in image generation since its proposal. Afterward, a vast effort has been devoted to exploring the applications, especially in text-to-image generation. GLIDE [34] leverages two different kinds of guidance, *i.e.*, classifier-free

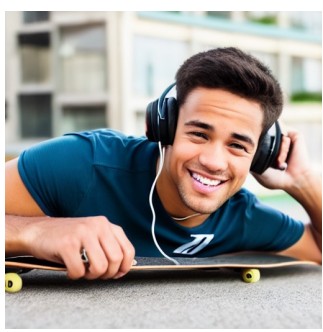

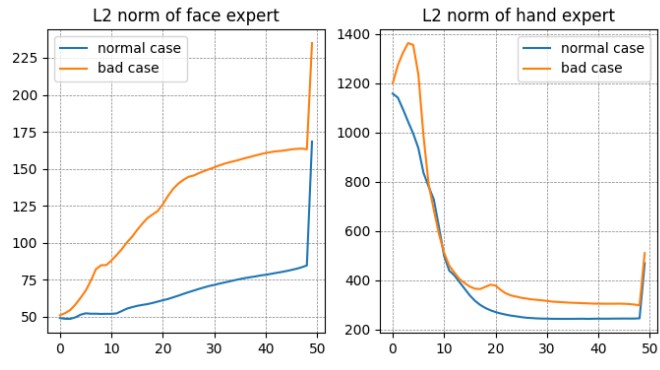

(a) An unrealistic case              (b) Averaged L2 norm of outputs

Figure 16: (a) is a showcase of a generated unrealistic image. In (b), the left is averaged L2 norm of outputs from face experts and the right is averaged L2 norm of outputs from hand experts. X is the timestep.

guidance [18] and clip guidance, to match the semantics of the generated image with the given text. Imagen [42] further improves the performance of text-to-image generation via a large T5 text encoder [37]. Stable Diffusion [40] uses a VAE encoder to map image to latent space and perform diffusion on representation. DALL-E 2 [38] transfers text representation encoded by CLIP [36] to image representation via diffusion prior. Besides generation, diffusion models have also been used in text-driven image editing [47]. Inspired by the key observation between text and map in the cross-attention module, Prompt-to-prompt [15] modifies the cross-attention map with prompt while preserving the original structure and content. Further Null-text inversion [31] achieves real image edition via image inversion. Different from them, our work primarily focuses on human-centric text-to-image generation, aiming to alleviate the poor performance of diffusion models in this field.

**Mixture-of-Experts.** MoE is first proposed in [20]. The underlying principle of MoE is that different subsets of data or contexts may be better modeled by distinct experts. Theoretically, MoE could scale model capability with little cost by using sparsely-gated MoE layer [44]. Researchers extend MoE on transformers by replacing FFN and attention layers with position-wise MoE layers [24, 8, 7]. Swith Transformer [11] simplifies the routing algorithm. [52] propose an Expert Choice (EC) routing algorithm to achieve optimal load balancing. GLaM [10] scales transformer model parameter to 1.2T but is inference-efficient. VMoE [39] scale vision model to 15B parameter via MoE. Soft MoE [35] introduces an implicit assignment by passing weighted combinations of all tokens to each expert. MoE has been adapted in generation tasks [12, 2]. For example, ERNIE-ViLG [12] uniformly divides the denoising process into several distinct stages, with each being associated with a specific model. eDiff-i [2] calculates thresholds to separate the whole process into three stages. Differing from employing experts in divided stages, we consider low-rank modules trained by customized datasets as experts to adaptively refine generation.

### A.12 More Discussion

**Dataset Contribution.** To the best of our knowledge, we newly collect and propose a very high-quality and comprehensive human-centric dataset simultaneously meeting legal compliance.

● *High-quality.* Our human-centric dataset consists of images of high resolution (basically over 1024 × 2048) and large file size, *e.g.*, 1M, 3M, 5M, and 10M, collected from websites (*e.g.*, https://www.pexels.com/search/people/ and https://unsplash.com/s/photos/people). The readers can simply click these links to have a look. Such quality is missing in current widely used image-text datasets, *e.g.*, LAION2b-en. To support it, we sample 350K human-centric images from LAION2b-en. The averaged height and width are approximately 455 and 415 respectively. Most of them are between 320 and 357. Compared to our human-centric dataset, they fail to provide sufficient prior and high-quality details.

●*Comprehensive.* As the natural face and hand are relatively hard to generate compared to other parts as discussed in the community, in addition of the collected human-in-the-scene subset, we also

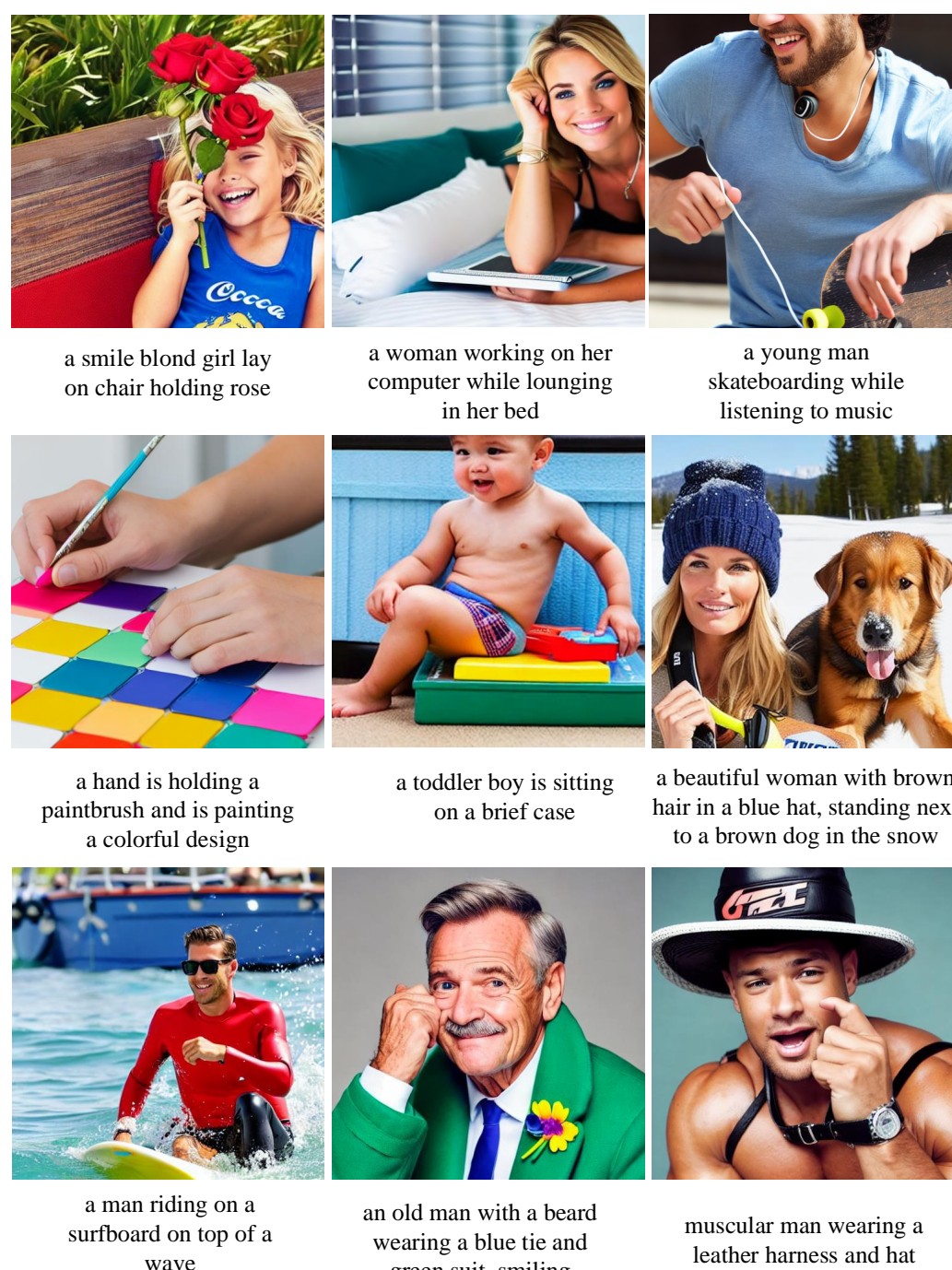

a smile blond girl lay on chair holding rose

a woman working on her computer while lounging in her bed

a young man skateboarding while listening to music

a hand is holding a paintbrush and is painting a colorful design

a toddler boy is sitting on a brief case

a beautiful woman with brown hair in a blue hat, standing next to a brown dog in the snow

a man riding on a surfboard on top of a wave

an old man with a beard wearing a blue tie and green suit, smiling

muscular man wearing a leather harness and hat

Figure 17: More images generated by MoLE. Zoom in for a better view.

provide two high-quality close-up of face and hand subsets. In particular, close-up images of hand are absent in the human-centric images sampled from LAION2b-en during our observations. To the best of our knowledge, this quantity of the high quality close-up hand images is absent in prior related studies. It will significantly help to address challenges associated with generating natural-looking hands and propel the advancement of human-centric image generation for subsequent researchers.

● *Legal Compliance.* The human-centric dataset is collected from websites including unsplash.com, gratisography.com, seeprettyface.com, morguefile.com, pexels.com, *etc*. Images on these websites are published by their respective authors under Public Domain CC0 1.0 3 license that allows free use, redistribution, and adaptation for non-commercial purposes. When collecting and filtering the data, we are careful to only include images that, to the best of our knowledge, are intended for free

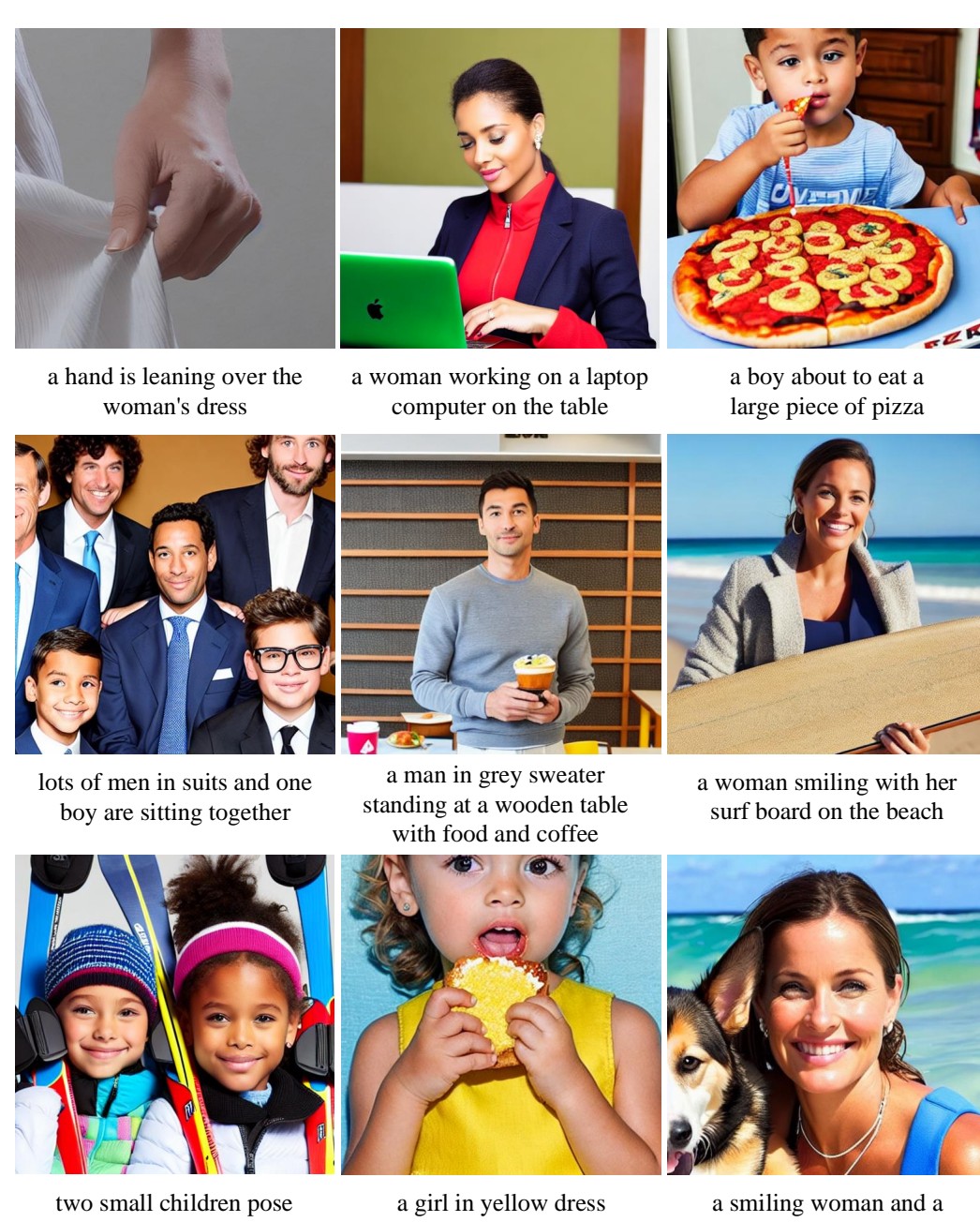

| | | |
|---|---|---|
| a hand is leaning over the woman's dress | a woman working on a laptop computer on the table | a boy about to eat a large piece of pizza |
| lots of men in suits and one boy are sitting together | a man in grey sweater standing at a wooden table with food and coffee | a woman smiling with her surf board on the beach |
| two small children pose together with skis and poles. | a girl in yellow dress eating a piece of cake | a smiling woman and a dog in the ocean |

Figure 18: More images generated by MoLE. Zoom in for a better view.

use. We are committed to protecting the privacy of individuals who do not wish their images to be included. The eventually resulted human-centric dataset can be used for academic purposes only and the users are required to ensure compliance with ethical and legal regulations.

**Analysis about Ratios of Different Races.** Since our dataset is collected from Internet, it inevitably inherits the race bias of target websites. Hence we provide an analysis about ratios of different races in MoLE with a single prompt "A beautiful woman". Specifically, we generate 10K images with this prompt. With the help of DeepFace, we find that approximately 51.08% individuals identify as White, 5.29% as Asian, 10.18% as Black, 4.31% as Indian, 24.66% as Latino Hispanic, and 4.48% as Middle Eastern. To verify if the generation of different races can be improved by using a race-balanced dataset, based on our dataset we use DeepFace to reconstruct a new dataset with the same ratio of different races. The newly curated dataset comprises 30k images. We use it to train a MoLE model and generate 10K images using the same prompt "A beautiful woman". We find that approximately 45.56% individuals identify as White, 7.17% as Asian, 8.32% as Black, 13.41% as Indian, 13.44%

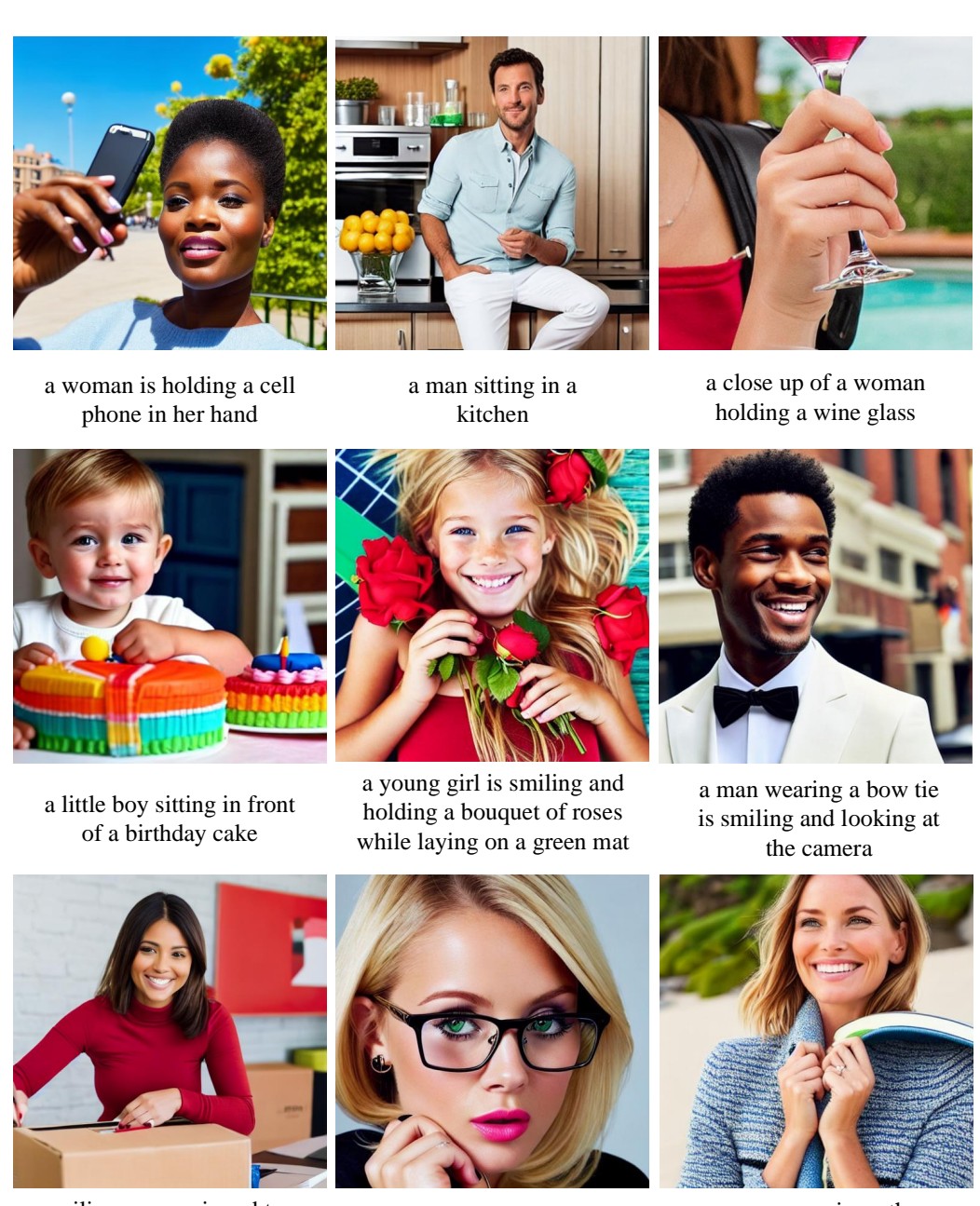

a woman is holding a cell phone in her hand

a man sitting in a kitchen

a close up of a woman holding a wine glass

a little boy sitting in front of a birthday cake

a young girl is smiling and holding a bouquet of roses while laying on a green mat

a man wearing a bow tie is smiling and looking at the camera

smiling woman in red top putting items in a box

a woman with blonde hair wearing glasses

a woman is on the beach, smiling

Figure 19: More images generated by MoLE. Zoom in for a better view.

as Latino Hispanic, and 12.10% as Middle Eastern. This result shows a relatively higher balance of races compared to previous one, demonstrating that MoLE is beneficial to alleviate race biases with the reconstructed dataset. Given the constraints of our race-balanced dataset, which contains only 30k samples, we believe that expanding the size of the race-balanced data could enhance our method's ability to further address the race-related issue.

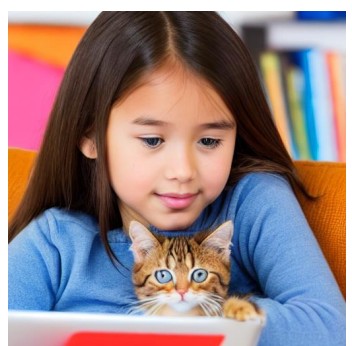
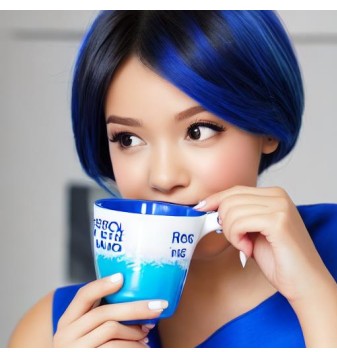
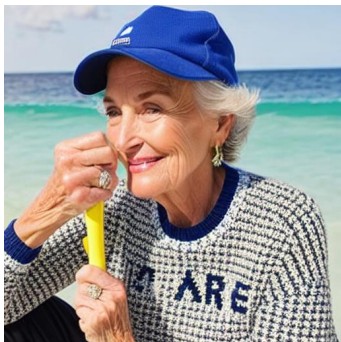

a girl sitting on a couch with a cat on top of her laptop

a woman with blue hair is drinking from a blue coffee cup

an older woman in a sweater sits at the beach

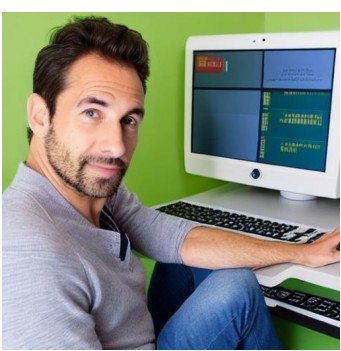
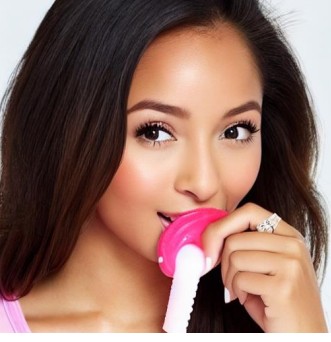
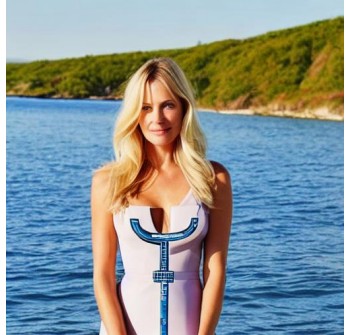

a man sitting in front of a large computer monitor

a woman with long brown hair is holding a pink toothbrush in her hand

a woman in a white dress stands on the edge of a lake

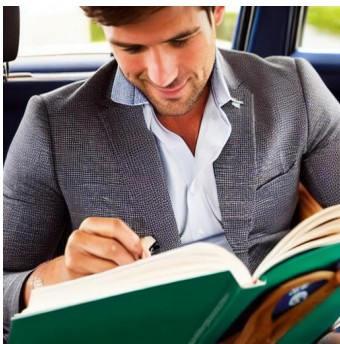
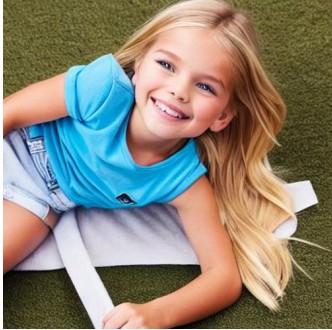
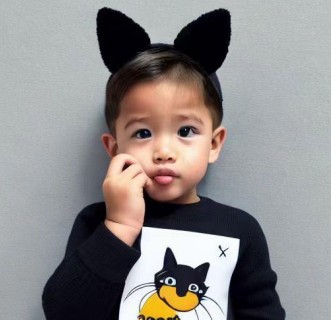

a man is sitting in a car reading a book

a pretty blonde haired girl wrapped up in a sheet and laying down

a young boy wearing a black shirt with a cat on it and cat ears

Figure 20: More images generated by MoLE. Zoom in for a better view.

The woman sits at the table overlooking the pink and white cake with lit candle.

A man and a woman standing behind a fruitstand of bananas at a chinese shop.

A sexy young blonde woman holding a tennis racquet.

A baby girl chews on a stick with a teddy bear in hand.

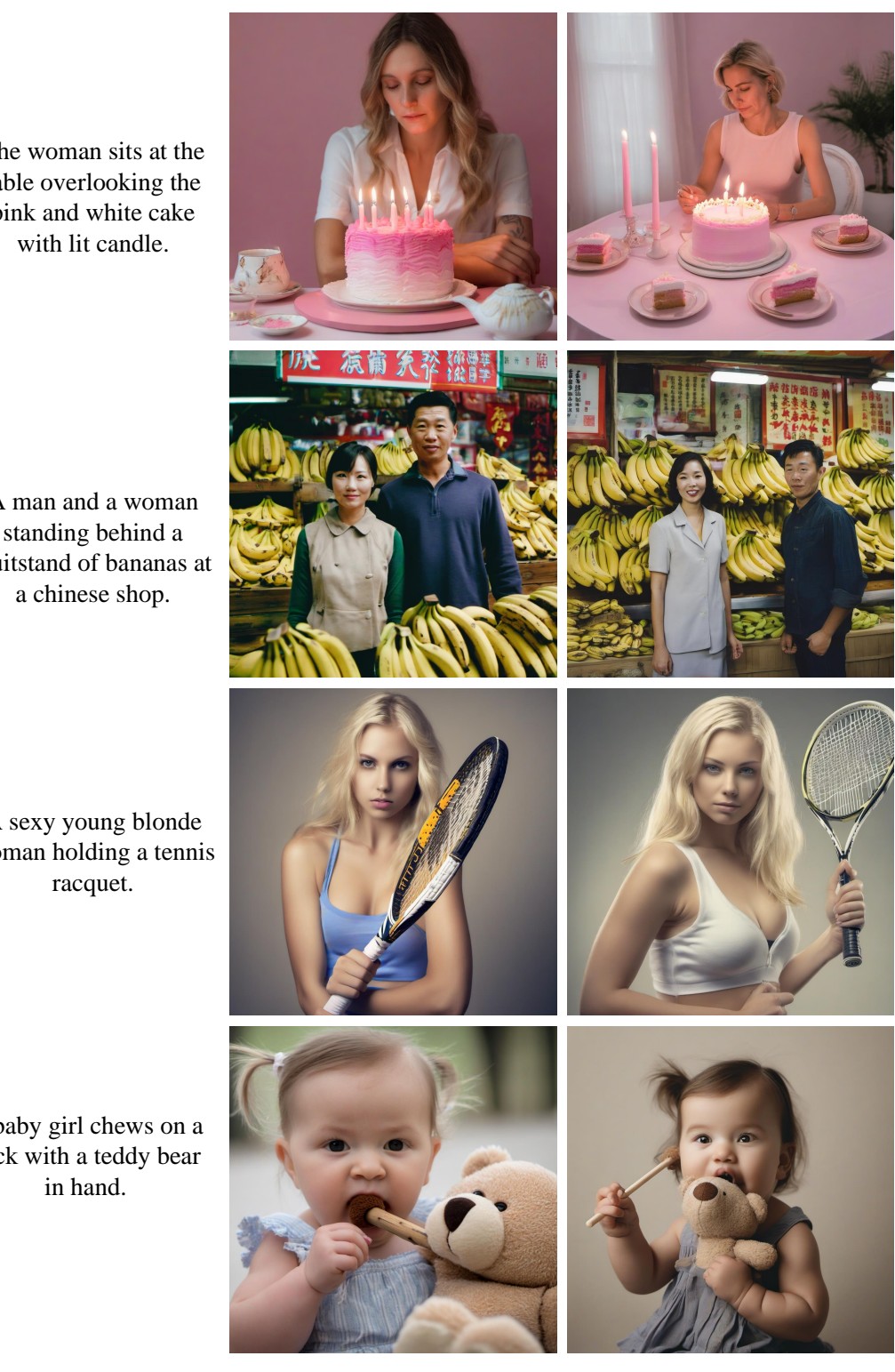

MoLE                    SDXL

Figure 21: Comparing MoLE with SDXL. Zoom in for a better view.

A woman wearing a white shirt holding the face of a white horse.

A woman lays on her side, as she wears beautiful, colorful clothing.

A man wearing a beret while using a laptop computer.

A girl eating a piece of cake from a pink princess plate

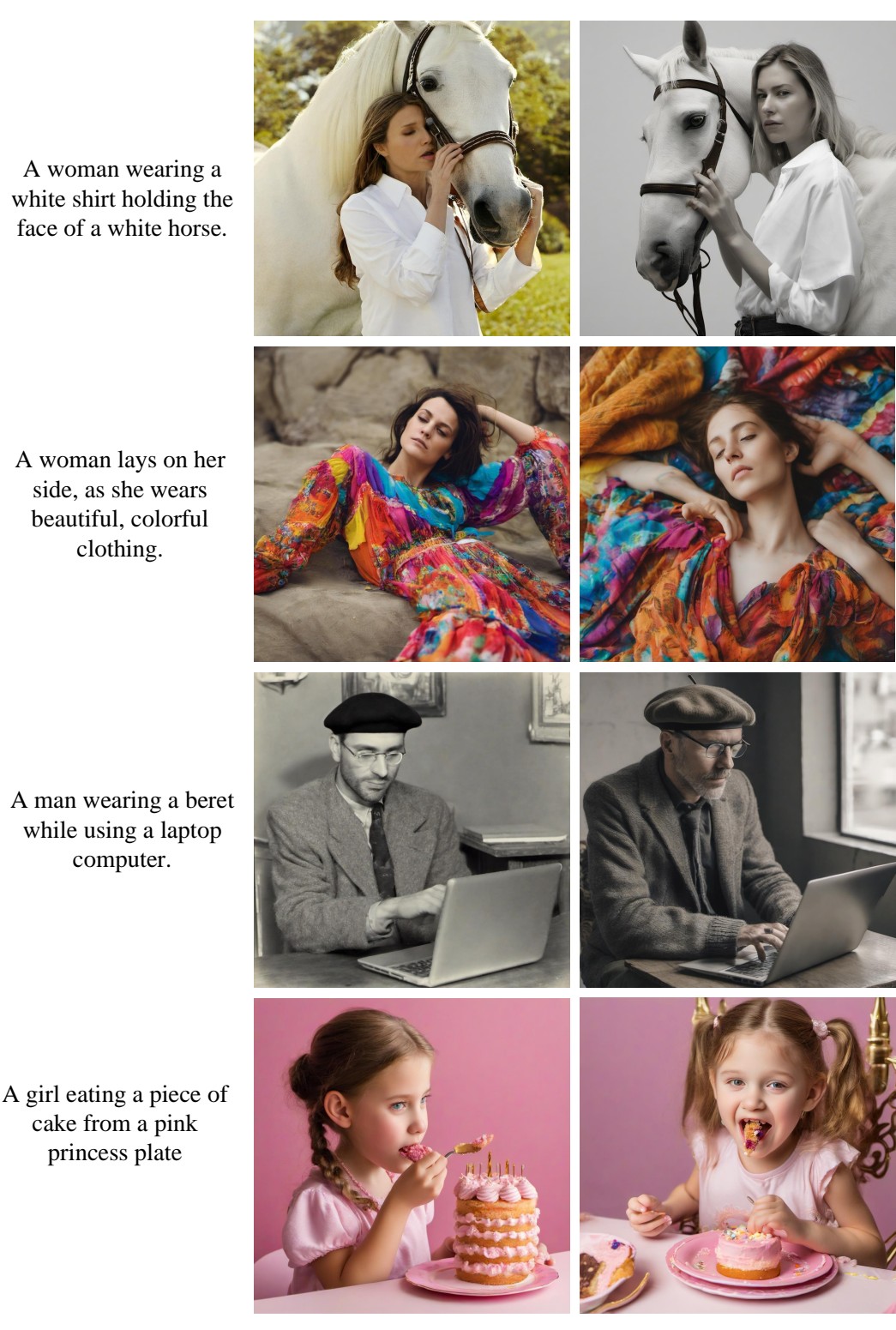

MoLE            SDXL

Figure 22: Comparing MoLE with SDXL. Zoom in for a better view.

A woman wearing glasses looking at slices of pizza

An elderly man is using a laptop computer at a desk.

A pregnant woman is in bed reading a large book.

A man dressed very nice posing for a picture.

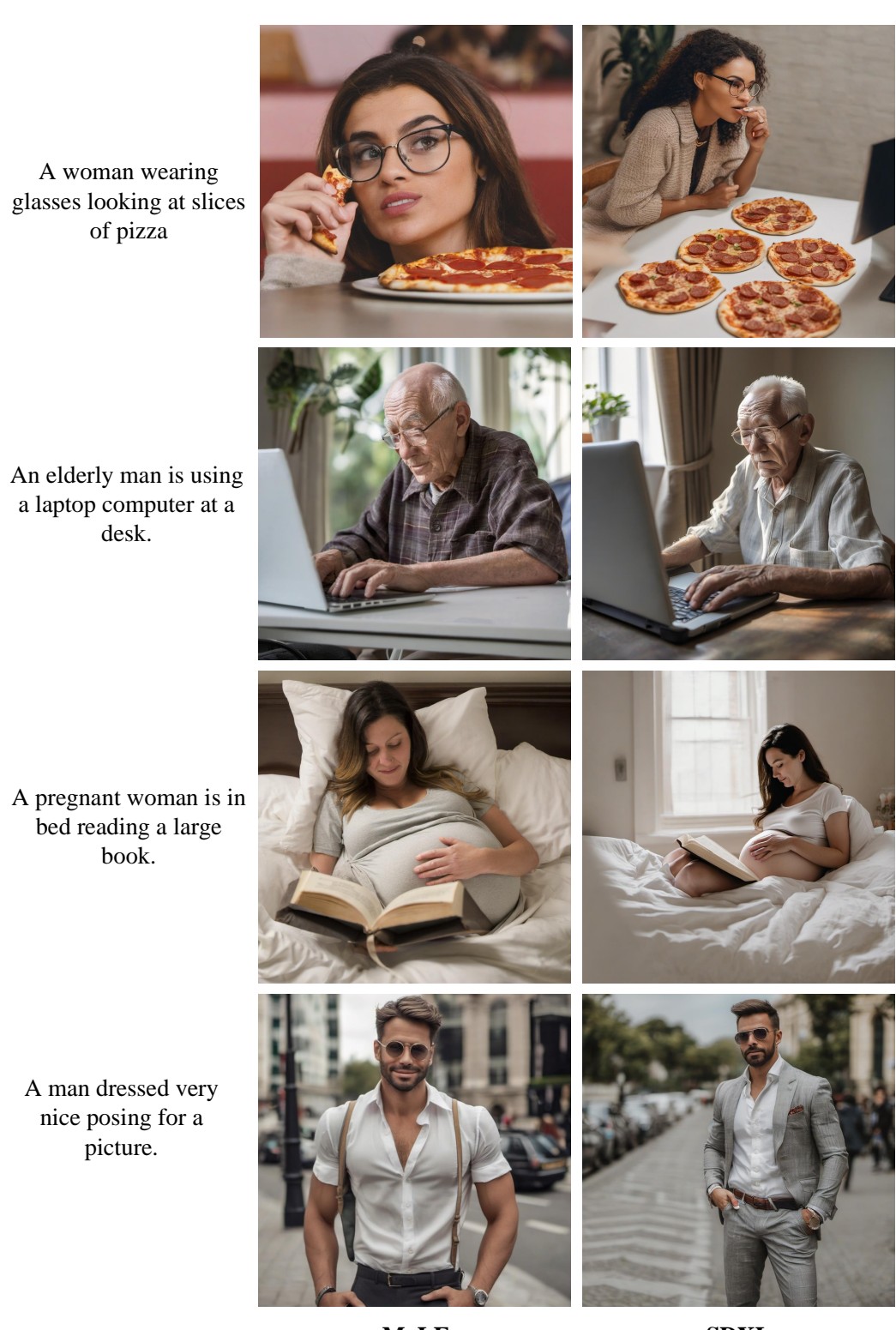

**MoLE**                    **SDXL**

Figure 23: Comparing MoLE with SDXL. Zoom in for a better view.

