# OpenReview forum: "MoLE: Enhancing Human-centric Text-to-image Diffusion via Mixture of  Low-rank Experts"
_NeurIPS.cc/2024/Conference — NeurIPS 2024 poster_

### Official Review · Reviewer_Rq6G · 2024-06-16

**Soundness:** 3
**Presentation:** 4
**Contribution:** 3
**Rating:** 7
**Confidence:** 5

**Summary:**

This paper aims to address the quality issues in human-centric text-to-image generation by constructing a large-scale dataset of millions of portraits. Additionally, the authors propose training experts for generating facial and hand details and efficiently integrate these experts using the MoE architecture into diffusion models. This integration enhances the realism of the final outputs in terms of facial and hand details.Extensive visualizations and analyses demonstrate the effectiveness of the proposed approach in this paper.

**Strengths:**

+This paper is well-written, clearly expressing the motivation and key contributions.

+The proposed Mixture of Low-rank Experts on text-to-image diffusion model framework is novel. To the best of my knowledge, this is the first work to try MoE learning strategy to enhance the human generation quality in diffusion models.

+The contributions are relatively substantial, especially with the author's collection of a publicly available high-quality human dataset. This dataset's availability will advance the community in deeper research into human generation.

+The visualized comparisons are comprehensive and demonstrate the advantages of the MoE architecture in diffusion models.

**Weaknesses:**

Overall, I am quite satisfied with this paper, but I still have some questions or concerns, as detailed below:


-- The text description process is inadequate, particularly regarding the manual filtering of data generated by the LLaVA model, which lacks any detailed description. As far as I know, LLaVA is trained on general image-text pairs, so there should be many hallucinated descriptions in the process of generating human image descriptions. How did the authors address this issue? Additionally, why did the authors not consider using some human/clothing attribute prediction methods and then use the generated attributes as prompts for the MLLM to make the results more reliable?

--The visualized comparison results presented by the authors are mostly half-body portraits. Could the authors show some full-body comparison results? In these images, the face and hand areas are smaller, increasing the probability of poor quality and making them more challenging to fix

**Questions:**

Most of my questions have been listed in the "weaknesses" section. Please respond to the above questions.

Additionally, to better evaluate its generalization capability, have the authors considered applying the MoE architecture to transformer-based diffusion models, such as PixArt[1]?

[1] Junsong Chen,  Jincheng Yu and etc. PixArt-$\alpha$: Fast Training of Diffusion Transformer for Photorealistic Text-to-Image Synthesis. ICLR, 2024.

**Limitations:**

See the "weakness" and "questions" section.

---

> ### Author Rebuttal · Authors · 2024-08-07
>
> We thank the reviewer for the positive comment, which is truly invigorating and encouraging! Below is our pointwise response, hoping to address your concerns.
>
> **Q1:** The text description process is inadequate, particularly regarding the manual filtering of data generated by the LLaVA model.
>
> **A1:** Sorry for the confusion. We take an image and a prompt "Describe this image in one sentence with details" as input for LLaVA to generate the caption of the image. Afterward, we will streamline the long LLaVA caption manually. Specifically, if the caption is very long, we will streamline it with a new shorter caption by ourselves while aligning with the content of the image. We also remove unrelated and uninformative text patterns including "creating ...", "demonstrating .....", etc. For example, if the caption contains, e.g., "creating a peaceful atmosphere", we will remove it to make the model focus on informative words more. We will add a detailed description of the process in our paper.
>
> **Q2:** How did the authors address the issue of hallucinated descriptions. Why did the authors not consider using some human/clothing attribute prediction methods and then use the generated attributes as prompts for the MLLM to make the results more reliable?
>
> **A2:** To alleviate this issue, we use CLIP to filter image-text pairs with lower scores. By doing so, we can effectively filter the hallucinated descriptions whose content does not appear in the image.
>
> For the second question, we also appreciate the reviewer's suggestion of using human/clothing attribute models. In our preliminary experiments, we qualitatively compared several CLIP-filtered LLava-generated captions with those generated by attribute models. Both approaches produced captions rich in detail and similar in their expression of global semantics. However, while the attribute model captions were more detailed due to the nature of these models, they occasionally appeared somewhat awkward and less natural compared to the normal descriptions of human habits. (For example,  "A full-body shot, an Asian adult female, outdoor, black straight above chest hair, a black silk shirt" **vs.** "an Asian adult female with black straight hair falling just above her chest, wearing a black silk shirt."). Another potential issue is that the attribute model may ignore the behavior of a person in an image, e.g., running and reading. Therefore, we opted not to use them in our paper, also considering that the former method was simpler to implement. Regardless, we appreciate the reviewer's valuable suggestion and will consider it to combine with our current method for future work.
>
> **Q3:** Could the authors show some full-body comparison results?
>
> **A3:** Yes, we show several full-body comparison results based on MoLE (SD v1.5) and MoLE (SDXL) in Fig.4 of the attached PDF file. One can see that even in the case of full-body images where the face and hand areas are smaller, MoLE still can refine these parts. We thank the reviewer's suggestion for this and will add this result to our paper.
>
> **Q4:** To better evaluate its generalization capability, have the authors considered applying the MoE architecture to transformer-based diffusion models, such as PixArt?
>
> **A4:**  We thank the reviewer's suggestion. To verify the generalization of our method, we attempt to build our MoLE based on PixArt-XL-2-512x512. To compare the performance, we randomly sample 3k prompts from COCO Human Prompts and calculate HPS for MoLE (PixArt) and PixArt.
> The evaluation process is repeated three times. Our method achieves $21.79 \pm 0.03$ HPS(%) and outperforms PixArt ($21.33 \pm 0.08$ HPS). These results demonstrate the generalization of our method. We are willing to cite and add the discussion to our paper.

---

> > ### Comment · Reviewer_Rq6G · 2024-08-11
> >
> > Thanks for the response. The response have addressed most of my concerns. Thus, I keep the "Accept" rating.

---

### Official Review · Reviewer_5LWQ · 2024-06-25

**Soundness:** 3
**Presentation:** 3
**Contribution:** 3
**Rating:** 6
**Confidence:** 5

**Summary:**

This paper enhances these Text-to-image diffusion models by introducing a curated dataset of over a million human-centric images and a novel method, MoLE, which utilizes specialized low-rank modules to improve facial and hand image quality in diffusion processes.
If the rebuilt dataset can be race-balanced and the approach could promote racial balance, I will be glad to raise my score.

**Strengths:**

1. The paper proposes a new human-centric dataset to enhance the human-centric generations, which is interesting.
2. Extensive experiments demonstrate the effectiveness of this approach.
3. The presentation is clear and easy to follow.

**Weaknesses:**

1. Stable Diffusion has introduced race biases when generating images. MoLE focuses on the generation qualities of faces and hands. If MoLE is beneficial to alleviate race biases with the reconstruction of the human-centric dataset, it will be more impactful like [1].

2. Limb deformation is also a big question in Stable Diffusion[2][3]. With the introduction of many human images featuring limbs in this dataset, it is crucial to assess whether this approach can effectively mitigate these deformations.

3. The generation of two global scalars may be tricky (Lines 180-182), current experiments show less evidence about the effects of the global scalars.

[1] ITI-GEN: Inclusive Text-to-Image Generation
[2] HumanSD: A Native Skeleton-Guided Diffusion Model for Human Image Generation.
[3] Towards Effective Usage of Human-Centric Priors in Diffusion Models for Text-based Human Image Generation.

**Questions:**

4. From lines 88-89, the main race is the white. Could the authors provide an analysis of the ratios of different races in one prompt (A Beautiful Woman)? Also, could the authors provide a discussion that if the ratios of different races are the same, will the generations of different races be improved?

5. The values of S in Figure 1 are more arbitrary to decide the qualities of a generation, so how to decide these values in MoLE?

6. Since the dataset is considered to be part of the innovation, will the dataset be released in the future?

**Limitations:**

The negative board impacts are insufficient in the Conclusion. The authors should analyze whether their approach will introduce more biases on race and other impacts, such as fake faces.

---

> ### Author Rebuttal · Authors · 2024-08-07
>
> We thank the reviewer for the encouraging comment! Below is our pointwise response, hoping to address your concerns.
>
> **Q1:** An analysis of the ratios of different races in one prompt (A Beautiful Woman)? Provide a discussion that if the ratios of different races are the same, will the generations of different races be improved?
>
> **A1:** We follow the reviewer's suggestion and use the prompt "A Beautiful Woman" to show the ratios of different races in MoLE. Specifically, we generate 10K images with this prompt. With the help of DeepFace (https://github.com/serengil/deepface), we find that approximately 51.08% individuals identify as white, 5.29% as Asian, 10.18% as Black, 4.31% as Indian, 24.66% as Latino Hispanic, and 4.48% as Middle Eastern.
>
> To verify if the generation of different races can be improved by using a race-balanced dataset, based on our dataset we use DeepFace to reconstruct a new dataset with the same ratio of different races. The newly created dataset comprises 30K images. We use it to train a MoLE model and generate 10K images using the same prompt "A Beautiful Woman". We find that approximately 45.56% individuals identify as white, 7.17% as Asian, 8.32% as Black, 13.41% as Indian, 13.44% as Latino Hispanic, and 12.10% as Middle Eastern. This result has a relatively higher balance of races compared to the previous result, demonstrating that MoLE is beneficial to alleviating race biases with the reconstructed dataset. We are willing to cite and add this discussion to our paper.
>
> **Q2:** With the introduction of many human images featuring limbs in this dataset, assess whether this approach can effectively mitigate limb deformations.
>
> **A2:** To assess whether MoLE can effectively mitigate limb deformations, we perform a user study by sampling 20 image pairs from SD v1.5 and MoLE, and inviting 50 participants to evaluate which model produces better human limbs with less deformations. Among the participants, we find that 62% of participants select MoLE, which indicates that MoLE also can effectively mitigate limb deformations.
>
> **Q3:** The generation of two global scalars may be tricky (Lines 180-182), current experiments show less evidence about the effects of the global scalars.
>
> **A3:** We feel sorry for the confusion. Actually, we have shown the efficacy of global scalars  in our ablation study of **Mixture Assignment** in Tab 3 where only employing global assignment can also improve the performance compared to SD v1.5 (as well as the model in Stage 1). Moreover, in (a) and (b) of Fig 8, we can see that global assignment is content-aware. For example, when generating a close-up image, e.g., a face image, the global assignment consciously produces large global scalars for the face expert and small global scalars for the hand expert. From this perspective, the global assignment is meaningful.
>
> **Q4:** The values of S in Figure 1 are more arbitrary to decide the qualities of a generation, so how to decide these values in MoLE?
>
> **A4:** In MoLE, we use a mechanism called **Soft Mixture Assignment** to determine these values. This mixture assignment contains two parts: global assignment and local assignment. Specifically, the global assignment takes as input the entire feature map to produce adaptive global scalers, which are allocated to each expert. Additionally, we introduce the local assignment, which takes as input each token to produce local scalers that similarly determine how much weight for each token sent to the experts.
>
> **Q5:** Since the dataset is considered to be part of the innovation, will the dataset be released in the future?
>
> **A5:**  Yes, we are very willing to release this dataset to advance the development of our community. Moreover, we promise to select a suitable license to ensure compliance with legal regulations and emphasize its exclusive use for academic purposes only.
>
> **Q6:** The negative board impacts are insufficient in the Conclusion. The authors should analyze whether their approach will introduce more biases on race and other impacts, such as fake faces.
>
> **A6:** We thank the reviewer's advice. Through our analysis in **A1** above, though MoLE may not introduce more biases on race, it also inherits the biases in the training data like pervious methods. As for other impacts such as fake faces, since our method primarily focuses on human-centric image generation, it also inevitably generates fake faces like other SD models, which requires users to leverage these generated images carefully and legally. These issues also warrant further research and consideration. We maintain transparency in our methods with open-source code and dataset composition, allowing for continuous improvement based on community feedback. We will highlight these discussions in the Broader Impact part.

---

> > ### Comment · Reviewer_5LWQ · 2024-08-08
> >
> > Thanks for the response. Most of my questions are well-discussed, but the answer to A1 shows that MoLE can only alleviate the bias a little even with a race-balanced dataset. So, I'll keep my score.

---

> > > ### Author Response · Authors · 2024-08-11
> > > **Response to Reviewer 5LWQ**
> > >
> > > We appreciate Reviewer 5LWQ's response. We think the limited effectiveness of MoLE in alleviating bias may be due to the vast size of the imbalanced training data (2 billion from LAION and 1 million from our own dataset), compared to the 30K samples in our race-balanced data, which constrains its impact. We believe that if we increase the amount of race-balanced data, our approach could further mitigate the race issue. We are actively working on collecting more race-balanced data and are working hard to alleviate this bias issue further.

---

### Official Review · Reviewer_NVnC · 2024-07-11

**Soundness:** 3
**Presentation:** 3
**Contribution:** 3
**Rating:** 7
**Confidence:** 3

**Summary:**

This paper aims to explore human-centric text-to-image generation, particularly in the context of faces and hands, the results often fall short of naturalness due to insufficient training priors. The authors alleviate the issue from two perspectives. 1) The authors collect a human-centric dataset with two specific sets of close-up images of faces and hands. 2) The authors propose Mixture of Low-rank Experts (MoLE) method by considering low-rank modules trained on close-up hand and face images respectively as experts.

**Strengths:**

1)This paper constructs a human-centric dataset comprising over one million high-quality human-in-the-scene images and two specific sets of close-up images of faces and hands. These datasets collectively provide a rich prior knowledge base to enhance the human-centric image generation capabilities of the diffusion model.

2)This paper proposes a simple yet effective method called Mixture of Low-rank Experts (MoLE) by considering low-rank modules trained on close-up hand and face images respectively as experts.

3)The paper is well-written and easy to follow. The dataset and benchmark of this paper are open source.

**Weaknesses:**

1) As the author mentioned, the soft mechanism is built on the fact that each token can adaptively determine how much (weight) should be sent to each expert by the sigmoid function. So, it will be better if the author can provide and analyze the distribution of the weight sent to each expert by the sigmoid function.

2) There are some minor writing errors in the paper, such as "is is" on line 208. The author needs to carefully check the manuscript.

**Questions:**

Please check the weakness.

**Limitations:**

The authors have discussed the limitations of the proposed dataset and method.

---

> ### Author Rebuttal · Authors · 2024-08-07
>
> We thank the reviewer for the positive comment! We hope our response presented below can address your concerns.
>
> **Q1:** Provide and analyze the distribution of the local weight sent to each expert by the sigmoid function.
>
> **A1:** We thank the reviewer's advice. We provide the distribution of the local weight sent to each expert in Fig.3 of the attached PDF file. To obtain this, we generate 10 samples for close-up images and normal human images, respectively, and collect local weights for each expert. In Fig.3, one can see that for close-up images, e.g., face, the corresponding expert receives more weights with higher values. We think this effectively demonstrates the efficacy of the soft assignment mechanism in MoLE, which adaptively activates the relevant expert to contribute more to the generation of close-up images. When generating normal human images involving face and hand, the two experts contribute equally, and generally, the face expert receives relatively more weights with higher values as the area of the face is typically larger than that of the hand. We will add this result and discussion in our paper.
>
> **Q2:** Some minor writing errors in the paper
>
> **A2:** Thanks for pointing out the typos. We will check our manuscript carefully and fix them.

---

> > ### Comment · Reviewer_NVnC · 2024-08-14
> >
> > Thanks for the response.
> >
> > My concerns are well-discussed, but considering the scores of other reviewers, I decide to keep my score.

---

### Official Review · Reviewer_AE6c · 2024-07-12

**Soundness:** 3
**Presentation:** 3
**Contribution:** 3
**Rating:** 5
**Confidence:** 4

**Summary:**

The authors propose a large-scale dataset for human image generation, comprising over one million images, including close-up face and hand subsets. They also introduce MoLE (Mixture of Low-rank Experts), a novel framework that utilizes two low-rank experts to learn face and hand representations. This approach shows promise for generating high-quality human images with precise control over face and hand features. Overall, the proposed dataset and method are a valuable contribution to the field of human image generation.

**Strengths:**

1. A large-scale human-centric dataset is proposed along with two close-up face and hand subsets.
2. A MoLE framework with two experts is proposed which is novel and interesting.

**Weaknesses:**

1. Comparison with existing datasets: While the proposed dataset is a significant contribution, a thorough comparison with established datasets like CosmicMan is essential to contextualize its value.
2. Comparison with state-of-the-art methods: The paper primarily focuses on generating realistic faces and hands, but it lacks a comprehensive comparison with relevant methods like HanDiffuser and HyperHuman. Comparing only with Stable Diffusion (SD) may not provide a complete picture. It would be beneficial to train existing methods on the proposed dataset and compare the results with the proposed method.
3. Ablation study on experts: It would be interesting to see the results of training only one expert compared to training two experts, to understand the impact of using multiple experts on the model's performance.

**Questions:**

My main concern is the comparison with existing datasets and methods. Could you provide more details about it?

**Limitations:**

Yes

---

> ### Author Rebuttal · Authors · 2024-08-07
>
> We thank the reviewer for the positive comment! Below is our pointwise response, hoping to address your concerns.
>
> **Q1:** Comparison with existing datasets like CosmicMan is essential to contextualize its value.
>
> **A1:** We thank the reviewer's advice. Below, we give a comparison of the differences between CosmicMan and our newly collected dataset, which primarily lie in four aspects:
>
> - From the aspect of image diversity, due to different motivations, CosmicMan only contains human-in-the-scene images while our dataset also involves two close-up datasets for face and hand, respectively. Moreover, to the best of our knowledge, the high-quality close-up hand dataset is absent in prior related studies.
>
> - From the aspect of image content distribution, there is a relatively severe gender imbalance in CosmicMan where female makes up a large proportion around 75% (Please see Fig 3 of Appendix in its paper) while our dataset is relatively balanced (58% vs 42%).
>
> - From the aspect of image size, though CosmicMan and our dataset are both of high quality, our collected images (basically over 1024 × 2048) are relatively larger than CosmicMan whose average size is 1488 × 1255.
>
> - From the aspect of data sources, our dataset is legally collected from various websites including unsplash.com, gratisography.com, morguefile.com, and pexels.com, etc, while CosmicMan is sourced from LAION-5B (See https://huggingface.co/datasets/cosmicman/CosmicManHQ-1.0 ). What sets our dataset apart is not just its wide collection, but also the freshness of the data. As a trade-off, the quantity of our dataset (1M) is relatively smaller than that of CosmicMan (5M).
>
> We will add this comparison to our paper to highlight the value of our dataset.
>
> **Q2:** Comparison with state-of-the-art methods like HanDiffuser and HyperHuman.
>
> **A2:** We thank the reviewer's advice. Regrettably, we find that the code for HanDiffuser (https://supreethn.github.io/research/handiffuser/index.html) and HyperHuman (https://github.com/snap-research/HyperHuman) has not been made available. As a result, we are unable to directly compare these methods with our work. We attempt to reimplement HyperHuman (it is relatively simpler) based on our understanding of the paper. However, due to our constraints in time and computational resources, we were unable to complete the reimplementation. Hence, we resort to a user study. Specifically, we invite 50 participants to compare the visualization presented in the two methods' papers with our generated images, respectively.  In the user study, we prepare 10 MoLE-HyperHuman pairs and ask participants to select the best one from each pair according to their preference in terms of hand quality. Some compared images are presented in Fig.1 and Fig.2 of the attached PDF file to show the differences between our generated images and theirs. The results show that 58% of participants think our generated images are better than that of HyperHuman. Similarly, for HanDiffuser, we also prepare 10 MoLE-HanDiffuser pairs and ask participants to select the best one. We find that 48% of participants vote for MoLE, slightly inferior to HanDiffuser (52%). All these results demonstrate that our method is effective and competitive with the state-of-the-art methods. More importantly, our method is user-friendly because both HanDiffuser and HyperHuman rely on additional conditions to enhance human and hand generation: HyperHuman takes text and skeleton as input; HanDiffuser needs text, a SMPL-H model, camera parameters, and hand skeleton. In contrast, MoLE only relies on text without the need for any additional conditions, offering greater flexibility and ease of use while maintaining competitive performance.
>
> **Q3:** Ablation study on experts to see the results of training only one expert compared to training two experts.
>
> **A3:** Following the reviewer's advice, we train only one expert and compare the performance with that of two experts. We find that one expert achieves $20.19 \pm 0.03$ HPS(%), inferior to that of two experts ($20.27 \pm 0.07$ HPS), which demonstrates the necessity of using one expert for face and hand, respectively.
>
> We hope that our responses above can address your concerns, and turn your assessment to the positive side. If you have any questions, please let us know during the rebuttal window. We appreciate your suggestions and comments!

---

### Author Rebuttal · Authors · 2024-08-07

## General Response

Thank all reviewers for their time and effort in reviewing our paper. We also thank all reviewers for their valuable feedback, which is instrumental in enhancing the quality of our work. We hope our pointwise responses below can clarify all reviewers’ confusion and alleviate all concerns.  **We add the visualization materials of our rebuttal in the attachment.**

Thank all reviewers’ time again.

---

### Decision · Program_Chairs · 2024-09-25

**Decision:**

Accept (poster)

**Comment:**

The submission proposes to improve text-to-image diffusion model performance when generating humans. The proposed approach involves training the model on a curated human-centric dataset, and training model experts on hand and face close-ups that are combined into an MoE model.

Reviewers felt that:
1. The paper was well-written and easy to follow
2. The contributions were substantial, including (i) the human-centric dataset; (ii) the MoE approach to improving generation performance; and (iii) the open-sourced project material.

At the same time, reviewers raised concerns about comparisons with existing datasets and methods, and potential ethics concerns, including wrt bias in the used datasets, and consent from subjects in the datasets. Post-rebuttal, all 4 reviewers (including the ethics reviewer) felt that these concerns were sufficiently addressed and recommend acceptance. The AC sees no reason to override the unanimous recommendation.

The authors are asked to include the recommendations of the reviewers (including the ethics reviewer) in the camera ready version.